# Differentially Private Federated Bayesian Optimization with Distributed Exploration

**Zhongxiang Dai**[†]**, Bryan Kian Hsiang Low**[†]**, Patrick Jaillet**[§]
Dept. of Computer Science, National University of Singapore, Republic of Singapore[†]
Dept. of Electrical Engineering and Computer Science, MIT, USA[§]
{daizhongxiang,lowkh}@comp.nus.edu.sg[†],jaillet@mit.edu[§]

## Abstract

*Bayesian optimization* (BO) has recently been extended to the *federated learning* (FL) setting by the *federated Thompson sampling* (FTS) algorithm, which has promising applications such as federated hyperparameter tuning. However, FTS is not equipped with a rigorous privacy guarantee which is an important consideration in FL. Recent works have incorporated *differential privacy* (DP) into the training of deep neural networks through a general framework for adding DP to iterative algorithms. Following this general DP framework, our work here integrates DP into FTS to preserve *user-level privacy*. We also leverage the ability of this general DP framework to handle different parameter vectors, as well as the technique of local modeling for BO, to further improve the utility of our algorithm through *distributed exploration* (DE). The resulting *differentially private FTS with DE* (DP-FTS-DE) algorithm is endowed with theoretical guarantees for both the privacy and utility and is amenable to interesting theoretical insights about the *privacy-utility trade-off*. We also use real-world experiments to show that DP-FTS-DE achieves high utility (competitive performance) with a strong privacy guarantee (small privacy loss) and induces a trade-off between privacy and utility.

## 1 Introduction

*Bayesian optimization* (BO) has become popular for optimizing expensive-to-evaluate black-box functions, such as tuning the hyperparameters of *deep neural networks* (DNNs) [56]. Motivated by the growing computational capability of edge devices and concerns over sharing the raw data, BO has recently been extended to the *federated learning* (FL) setting [46] to derive the setting of *federated BO* (FBO) [12]. The FBO setting allows multiple agents with potentially heterogeneous objective functions to collaborate in black-box optimization tasks without requiring them to share their raw data. For example, mobile phone users can use FBO to collaborate in optimizing the hyperparameters of their DNN models used in a smart keyboard application without sharing their sensitive raw data. Hospitals can use FBO to collaborate with each other when selecting the patients to perform a medical test [67] without sharing the sensitive patient information. An important consideration in FL has been a rigorous protection of the privacy of the users/agents, i.e., how to guarantee that by participating in a FL system, an agent would not reveal sensitive information about itself [29]. Furthermore, incorporating rigorous privacy preservation into BO has recently attracted increasing attention due to its importance to real-world BO applications [33, 36, 48, 71]. However, the state-of-the-art algorithm in the FBO setting, *federated Thompson sampling* (FTS) [12], is not equipped with privacy guarantee and thus lacks rigorous protection of the sensitive agent information.

*Differential privacy* (DP) [20] provides a rigorous privacy guarantee for data release and has become the state-of-the-art method for designing privacy-preserving ML algorithms [28]. Recently, DP has been applied to the iterative training of DNNs using *stochastic gradient descent* (DP-SGD) [1]

and the FL algorithm of *federated averaging* (DP-FedAvg) [47], which have achieved competitive performances (utility) with a strong privacy guarantee. Notably, these methods have followed a general framework for adding DP to generic iterative algorithms [45] (referred to as *the general DP framework* hereafter), which applies a *subsampled Gaussian mechanism* (Sec. 2) in every iteration. For an iterative algorithm (e.g., FedAvg) applied to a database with multiple *records* (e.g., data from multiple users), the general DP framework [45] can hide the participation of any single record in the algorithm in a principled way. For example, DP-FedAvg [47] guarantees (with high probability) that an adversary, even with arbitrary side information, cannot infer whether the data from a particular user has been used by the algorithm, hence preserving *user-level privacy*. Unfortunately, FTS [12] is not amenable to a straightforward integration of the general DP framework [45] (Sec. 3.1). So, we modify FTS to be compatible with the general DP framework and hence introduce the DP-FTS algorithm to preserve user-level privacy in the FBO setting. In addition to the theoretical challenge of accounting for the impact of the integration of DP in our theoretical analysis, we have to ensure that DP-FTS preserves the practical performance advantage (utility) of FTS. To this end, we leverage the ability of the general DP framework to handle different parameter vectors [45], as well as the method of local modeling for BO, to further improve the practical performance (utility) of DP-FTS.

Note that FTS, as well as DP-FTS, is able to achieve better performance (utility) than standard TS by *accelerating exploration* using the information from the other agents (aggregated by the central server) [12]. That is, an agent using FTS/DP-FTS benefits from needing to perform less exploration *in the early stages*. To improve the utility of DP-FTS even more, we further accelerate exploration in the early stages using our proposed *distributed exploration* technique which is a combination of local modeling for BO and the ability of the general DP framework to handle different parameter vectors. Specifically, we divide the entire search space into smaller local *sub-regions* and let every agent explore only one local sub-region *at initialization*. As a result, compared with the entire search space, every agent can explore the local sub-region more effectively because its *Gaussian process* (GP) surrogate (i.e., the surrogate used by BO to model the objective function) can model the objective function more accurately in a smaller local sub-region [21]. Subsequently, in every BO iteration, the central server aggregates the information (vector) for every sub-region separately: For a sub-region, the aggregation (i.e., weighted average) gives more emphasis (i.e., weights) to the information (vectors) from those agents who are assigned to explore this particular sub-region. Interestingly, this technique can be seamlessly integrated into the general DP framework due to its ability to process different parameter vectors (i.e., one vector for every sub-region) while still preserving the interpretation as a single subsampled Gaussian mechanism [45] (Sec. 3.3).[1] As a result, the information aggregated by the central server can help the agents explore every sub-region (hence the entire search space) more effectively in the early stages and thus significantly improve the practical convergence (utility), as demonstrated in our experiments (Sec. 5). We refer to the resulting DP-FTS algorithm with *distributed exploration* (DE) as DP-FTS-DE. Note that DP-FTS is a special case of DP-FTS-DE with only one sub-region (i.e., entire search space). So, we will refer to DP-FTS-DE as our main algorithm in the rest of this paper.

In this paper, we introduce the *differentially private FTS with DE* (DP-FTS-DE) algorithm (Sec. 3), the first algorithm with a rigorous guarantee on the user-level privacy in the FBO setting. DP-FTS-DE guarantees that an adversary cannot infer whether an agent has participated in the algorithm, hence assuring every agent that its participation will not reveal its sensitive information.[2] We provide theoretical guarantees for both the privacy and utility of DP-FTS-DE, which combine to yield a number of elegant theoretical insights about the *privacy-utility trade-off* (Sec. 4). Next, we empirically demonstrate that DP-FTS-DE delivers an effective performance with a strong privacy guarantee and induces a favorable trade-off between privacy and utility in real-world applications (Sec. 5).

## 2 Background

**Bayesian optimization (BO).** BO aims to maximize an objective function $f$ on a domain $\mathcal{X} \subset \mathbb{R}^D$ through sequential queries, i.e., to find $\mathbf{x}^* \in \arg\max_{\mathbf{x} \in \mathcal{X}} f(\mathbf{x})$.[3] Specifically, in iteration $t \in [T]$ (we use $[N]$ to represent $\{1, \ldots, N\}$ for brevity), an input $\mathbf{x}_t \in \mathcal{X}$ is selected to be queried to yield

---

[1] By analogy, the vectors for different sub-regions in our algorithm play a similar role to the parameters of different layers of a DNN in DP-FedAvg [47].

[2] Following [47], we assume that the central server is trustworthy and that the clients are untrustworthy.

[3] For simplicity, we assume $\mathcal{X}$ to be discrete, but our theoretical analysis can be extended to compact domains through suitable discretizations [7].

an output observation: $y_t \triangleq f(\mathbf{x}_t) + \zeta$ where $\zeta$ is sampled from a Gaussian noise with variance $\sigma^2$: $\zeta \sim \mathcal{N}(0, \sigma^2)$. To select the $\mathbf{x}_t$'s intelligently, BO uses a *Gaussian process* (GP) [54] as a surrogate to model the objective function $f$. A GP is defined by its mean function $\mu$ and kernel function $k$. We assume w.l.o.g. that $\mu(\mathbf{x}) = 0$ and $k(\mathbf{x}, \mathbf{x}') \leq 1, \forall \mathbf{x}, \mathbf{x}' \in \mathcal{X}$. We mainly focus on the widely used *squared exponential* (SE) kernel in this work. In iteration $t + 1$, given the first $t$ input-output pairs, the GP posterior is given by $\mathcal{GP}(\mu_t(\cdot), \sigma_t^2(\cdot, \cdot))$ where

$$\mu_t(\mathbf{x}) \triangleq \mathbf{k}_t(\mathbf{x})^\top (\mathbf{K}_t + \lambda \mathbf{I})^{-1} \mathbf{y}_t \,, \sigma_t^2(\mathbf{x}, \mathbf{x}') \triangleq k(\mathbf{x}, \mathbf{x}') - \mathbf{k}_t(\mathbf{x})^\top (\mathbf{K}_t + \lambda \mathbf{I})^{-1} \mathbf{k}_t(\mathbf{x}') \qquad (1)$$

in which $\mathbf{k}_t(\mathbf{x}) \triangleq (k(\mathbf{x}, \mathbf{x}_{t'}))_{t' \in [t]}^\top$, $\mathbf{y}_t \triangleq (y_{t'})_{t' \in [t]}^\top$, $\mathbf{K}_t \triangleq (k(\mathbf{x}_{t'}, \mathbf{x}_{t''}))_{t', t'' \in [t]}$ and $\lambda > 0$ is a regularization parameter [7]. In iteration $t + 1$, the GP posterior (1) is used to select the next query $\mathbf{x}_{t+1}$. For example, the *Thompson sampling* (TS) [7] algorithm firstly samples a function $f_{t+1}$ from the GP posterior (1) and then chooses $\mathbf{x}_{t+1} = \arg \max_{\mathbf{x} \in \mathcal{X}} f_{t+1}(\mathbf{x})$. BO algorithms are usually analyzed in terms of *regret*. A hallmark for well-performing BO algorithms is to be asymptotically *no-regret*, which requires the *cumulative regret* $R_T \triangleq \sum_{t=1}^T (f(\mathbf{x}^*) - f(\mathbf{x}_t))$ to grow sub-linearly.

*Random Fourier features* (RFFs) [53] has been adopted to approximate the kernel function $k$ using $M$-dimensional random features $\boldsymbol{\phi}$: $k(\mathbf{x}, \mathbf{x}') \approx \boldsymbol{\phi}(\mathbf{x})^\top \boldsymbol{\phi}(\mathbf{x}')$. RFFs offers a high-probability guarantee on the approximation quality: $\sup_{\mathbf{x}, \mathbf{x}' \in \mathcal{X}} |k(\mathbf{x}, \mathbf{x}') - \boldsymbol{\phi}(\mathbf{x})^\top \boldsymbol{\phi}(\mathbf{x}')| \leq \varepsilon$ where $\varepsilon = \mathcal{O}(M^{-1/2})$ [53]. Of note, RFFs makes it particularly convenient to approximately sample a function from the GP posterior. Specifically, define $\boldsymbol{\Phi}_t \triangleq (\boldsymbol{\phi}(\mathbf{x}_{t'}))_{t' \in [t]}^\top$ (i.e., a $t \times M$-dimensional matrix), $\boldsymbol{\Sigma}_t \triangleq \boldsymbol{\Phi}_t^\top \boldsymbol{\Phi}_t + \lambda \mathbf{I}$, and $\boldsymbol{\nu}_t \triangleq \boldsymbol{\Sigma}_t^{-1} \boldsymbol{\Phi}_t^\top \mathbf{y}_t$. To sample a function $\widetilde{f}$ from approximate GP posterior, we only need to sample

$$\boldsymbol{\omega} \sim \mathcal{N}(\boldsymbol{\nu}_t, \lambda \boldsymbol{\Sigma}_t^{-1}) \qquad (2)$$

and set $\widetilde{f}(\mathbf{x}) = \boldsymbol{\phi}(\mathbf{x})^\top \boldsymbol{\omega}, \forall \mathbf{x} \in \mathcal{X}$. RFFs has been adopted by FTS [12] since it allows avoiding the sharing of raw data and improves the communication efficiency [12].

**Federated Bayesian Optimization (FBO).** FBO involves $N$ agents $\mathcal{A}_1, \ldots, \mathcal{A}_N$. Every agent $\mathcal{A}_n$ attempts to maximize its objective function $f^n : \mathcal{X} \to \mathbb{R}$, i.e., to find $\mathbf{x}^{n,*} \in \arg \max_{\mathbf{x} \in \mathcal{X}} f^n(\mathbf{x})$, by querying $\mathbf{x}_t^n$ and observing $y_t^n, \forall t \in [T]$. Without loss of generality, our theoretical analyses mainly focus on the perspective of agent $\mathcal{A}_1$. That is, we derive an upper bound on the cumulative regret of $\mathcal{A}_1$: $R_T^1 \triangleq \sum_{t=1}^T (f^1(\mathbf{x}^{1,*}) - f^1(\mathbf{x}_t^1))$ in Sec. 4. We characterize the similarity between $\mathcal{A}_1$ and $\mathcal{A}_n$ by $d_n \triangleq \max_{\mathbf{x} \in \mathcal{X}} |f^1(\mathbf{x}) - f^n(\mathbf{x})|$ such that $d_1 = 0$ and a smaller $d_n$ indicates a larger degree of similarity between $\mathcal{A}_1$ and $\mathcal{A}_n$. Following the work of [12], we assume that all participating agents share the same set of random features $\boldsymbol{\phi}(\mathbf{x}), \forall \mathbf{x} \in \mathcal{X}$. In our theoretical analysis, we assume that all objective functions have a bounded norm induced by the *reproducing kernel Hilbert space* (RKHS) associated with the kernel $k$, i.e., $\|f^n\|_k \leq B, \forall n \in [N]$. The work of [12] has introduced the FTS algorithm for the FBO setting. In iteration $t + 1$ of FTS, every agent $\mathcal{A}_n$ $(2 \leq n \leq N)$ samples a vector $\boldsymbol{\omega}_{n,t}$ from its GP posterior (2) and sends it to $\mathcal{A}_1$. Next, with probability $p_t \in (0, 1]$, $\mathcal{A}_1$ chooses the next query using a function $f_{t+1}^1$ sampled from its own GP posterior: $\mathbf{x}_{t+1} = \arg \max_{\mathbf{x} \in \mathcal{X}} f_{t+1}^1(\mathbf{x})$; with probability $1 - p_t$, $\mathcal{A}_1$ firstly randomly samples an agent $\mathcal{A}_n$ $(2 \leq n \leq N)$ and then chooses $\mathbf{x}_{t+1} = \arg \max_{\mathbf{x} \in \mathcal{X}} \boldsymbol{\phi}(\mathbf{x})^\top \boldsymbol{\omega}_{n,t}$.

**Differential Privacy (DP).** DP provides a rigorous framework for privacy-preserving data release [19]. Consistent with that of [47], we define two datasets as *adjacent* if they differ by the data of a single user/agent, which leads to the definition of user-level DP:

**Definition 1.** *A randomized mechanism* $\mathcal{M} : \mathcal{D} \to \mathcal{R}$ *satisfies* $(\epsilon, \delta)$-*DP if for any two adjacent datasets* $D_1$ *and* $D_2$ *and any subset of outputs* $\mathcal{S} \subset \mathcal{R}$, $\mathbb{P}(\mathcal{M}(D_1) \in \mathcal{S}) \leq e^\epsilon \mathbb{P}(\mathcal{M}(D_2) \in \mathcal{S}) + \delta$ .

Here, $\epsilon$ and $\delta$ are DP parameters such that the smaller they are, the better the privacy guarantee. Intuitively, user-level DP ensures that adding or removing any single user/agent has an imperceptible impact on the output of the algorithm. DP-FedAvg [47] has added user-level DP into the FL setting by adopting a general DP framework [45]. In DP-FedAvg, the central server applies a *subsampled Gaussian mechanism* to the vectors (gradients) received from multiple agents in every iteration $t$:

1. Select a subset of agents by choosing every agent with a fixed probability $q$,
2. Clip the vector $\boldsymbol{\omega}_{n,t}$ from every selected agent $n$ so that its $L_2$ norm is upper-bounded by $S$,
3. Add Gaussian noise (variance proportional to $S^2$) to the weighted average of clipped vectors.

The central server then broadcasts the vector produced by step 3 to all agents. As a result of the general DP framework, they are able to not only provide a rigorous privacy guarantee, but also use the *moments accountant* method [1] to calculate the *privacy loss*.[4] More recently, the work of [45] has formalized the methods used by [1] and [47] as a general DP framework that is applicable to generic iterative algorithms. Notably, the general DP framework [45] can naturally process different parameter vectors (e.g., parameters of different layers of a DNN), which is an important property that allows us to integrate distributed exploration (Sec. 3.2) into our algorithm.

# 3 Differentially Private FTS with Distributed Exploration

Here we firstly introduce how we modify FTS to integrate DP to derive the DP-FTS algorithm (Sec. 3.1). Then, we describe distributed exploration which can be seamlessly integrated into DP-FTS to further improve utility (Sec. 3.2). Lastly, we present the complete DP-FTS-DE algorithm (Sec. 3.3).

## 3.1 Differentially Private Federated Thompson Sampling (DP-FTS)

The original FTS algorithm [12] is not amenable to a straightforward integration of the general DP framework. This is because FTS [12] requires an agent to receive all vectors $\boldsymbol{\omega}_{n,t}$'s from the other agents (Sec. 2), so the transformations to the vectors required by the general DP framework (e.g., subsampling and weighted averaging of the vectors) cannot be easily incorporated. Therefore, we modify FTS by (a) adding a central server for performing the privacy-preserving transformations, and (b) passing a single aggregated vector (instead of one vector from every agent) to the agents. Fig. 1a illustrates our DP-FTS algorithm which is obtained by integrating the general DP framework into modified FTS. Every iteration $t$ of DP-FTS consists of the following steps:

① , ② **(by Agents):** Every agent $\mathcal{A}_n$ samples a vector $\boldsymbol{\omega}_{n,t}$ (2) using its own current history of $t$ input-output pairs (step ①) and sends $\boldsymbol{\omega}_{n,t}$ to the central server (step ②).

③ , ④ **(by Central Server):** Next, the central server processes the $N$ received vectors $\boldsymbol{\omega}_{n,t}$ using a subsampled Gaussian mechanism (step ③): It (a) selects a subset of agents $\mathcal{S}_t$ by choosing each agent with probability $q$, (b) clips the vector $\boldsymbol{\omega}_{n,t}$ of every selected agent s.t. its $L_2$ norm is upper-bounded by $S$, and (c) calculates a *weighted average* of the clipped vectors using weights $\{\varphi_n, \forall n \in [N]\}$, and adds to it a Gaussian noise. The final vector $\boldsymbol{\omega}_t$ is then broadcast to all agents (step ④).

⑤ **(by Agents):** After an agent $\mathcal{A}_n$ receives the vector $\boldsymbol{\omega}_t$ from the central server, it can choose the next query $\mathbf{x}_{t+1}^n$ (step ⑤): With probability $p_{t+1} \in (0,1]$, $\mathcal{A}_n$ chooses $\mathbf{x}_{t+1}^n$ using standard TS by firstly sampling a function $f_{t+1}^n$ from its own GP posterior of $\mathcal{GP}(\mu_t^n(\cdot), \beta_{t+1}^2 \sigma_t^n(\cdot,\cdot)^2)$ (1) where $\beta_t \triangleq B + \sigma\sqrt{2(\gamma_{t-1} + 1 + \log(4/\delta))}$,[5] and then choosing $\mathbf{x}_{t+1}^n = \arg\max_{\mathbf{x} \in \mathcal{X}} f_{t+1}^n(\mathbf{x})$; with probability $1 - p_{t+1}$, $\mathcal{A}_n$ chooses $\mathbf{x}_{t+1}^n$ using $\boldsymbol{\omega}_t$ received from the central server: $\mathbf{x}_{t+1}^n = \arg\max_{\mathbf{x} \in \mathcal{X}} \phi(\mathbf{x})^\top \boldsymbol{\omega}_t$. Consistent with that of [12], $(p_t)_{t \in \mathbb{Z}^+}$ is chosen as a monotonically increasing sequence such that $p_t \in (0,1], \forall t$ and $p_t \to 1$ as $t \to \infty$. This choice of the sequence of $(p_t)_{t \in \mathbb{Z}^+}$ ensures that in the early stage (i.e., when $1 - p_t$ is large), an agent can leverage the information from the other agents (via $\boldsymbol{\omega}_t$) to improve its convergence by accelerating its exploration.

After choosing $\mathbf{x}_{t+1}^n$ and observing $y_{t+1}^n$, $\mathcal{A}_n$ adds $(\mathbf{x}_{t+1}^n, y_{t+1}^n)$ to its history and samples a new vector $\boldsymbol{\omega}_{n,t+1}$ (step ①). Next, $\mathcal{A}_n$ sends $\boldsymbol{\omega}_{n,t+1}$ to the central server (step ②), and the algorithm is repeated. The detailed algorithm will be presented in Sec. 3.3 since DP-FTS is equivalent to the DP-FTS-DE algorithm with $P = 1$ sub-region.

## 3.2 Distributed Exploration (DE)

To accelerate the practical convergence (utility) of DP-FTS, we introduce DE to further accelerate the exploration in the early stages (Sec. 1). At the beginning of BO, a small number of initial points are usually selected from the entire domain using an exploration method (e.g., random search) to warm-start BO. We use DE to allow every agent to explore a smaller local sub-region *at initialization*, which is easier to model for the GP surrogate [21], and leverage the ability of the general DP framework to handle different parameter vectors [45] to integrate DE into DP-FTS in a seamless way.

---

[4]Given a $\delta$, the privacy loss is defined as an upper bound on $\epsilon$ calculated by the moments accountant method.
[5]$\gamma_{t-1}$ is the max information about the objective function from any $t-1$ queries and $\delta \in (0,1)$ (Theorem 1).

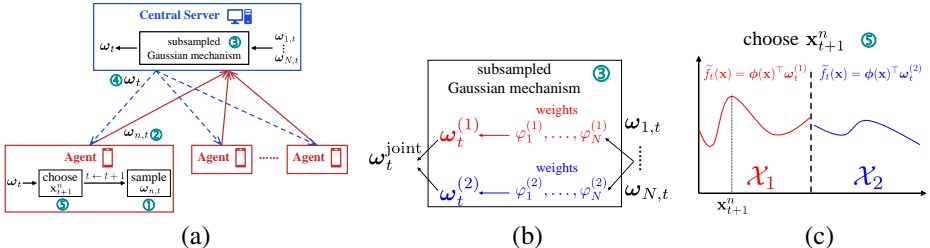

Figure 1: (a) DP-FTS algorithm (without distributed exploration). (b-c) Replacing steps ③ and ⑤ in (a) with that in (b) and (c) respectively to derive the DP-FTS-DE algorithm ($P = 2$).

Specifically, we partition the input domain $\mathcal{X}$ into $P \geq 1$ disjoint sub-regions: $\mathcal{X}_1, \ldots, \mathcal{X}_P$ such that $\cup_{i=1,\ldots,P} \mathcal{X}_i = \mathcal{X}$ and $\mathcal{X}_i \cap \mathcal{X}_j = \emptyset, \forall i \neq j$. At initialization, we assign every agent $\mathcal{A}_n$ to explore (i.e., choose the initial points randomly from) a particular sub-region $\mathcal{X}_{i_n}$.[6] Note that if an agent $\mathcal{A}_n$ is assigned to explore a sub-region $\mathcal{X}_{i_n}$ (instead of exploring the entire domain), its vector $\boldsymbol{\omega}_{n,t}$ (2) sent to the central server is more informative about its objective function *in this sub-region* $\mathcal{X}_{i_n}$.[7] As a result, for a sub-region $\mathcal{X}_i$, the vectors from those agents exploring $\mathcal{X}_i$ contain information that can help better explore $\mathcal{X}_i$. So, we let the central server construct a separate vector $\boldsymbol{\omega}_t^{(i)}$ for every sub-region $\mathcal{X}_i$ and when constructing $\boldsymbol{\omega}_t^{(i)}$, give more weights to the vectors from those agents exploring $\mathcal{X}_i$ because they are more informative about $\mathcal{X}_i$. Consequently, the central server needs to construct $P$ different vectors $\{\boldsymbol{\omega}_t^{(i)}, \forall i \in [P]\}$ with each $\boldsymbol{\omega}_t^{(i)}$ using a separate set of weights $\{\varphi_n^{(i)}, \forall n \in [N]\}$. Interestingly, from the perspective of the general DP framework [45], the different vectors $\{\boldsymbol{\omega}_t^{(i)}, \forall i \in [P]\}$ can be interpreted as analogous to the parameters of different layers of a DNN and can thus be naturally handled by the general DP framework. After receiving the $P$ vectors from the central server, every agent uses $\boldsymbol{\omega}_t^{(i)}$ to reconstruct the sampled function in the sub-region $\mathcal{X}_i$: $\widetilde{f}_t(\mathbf{x}) = \boldsymbol{\phi}(\mathbf{x})^\top \boldsymbol{\omega}_t^{(i)}, \forall \mathbf{x} \in \mathcal{X}_i$, and then (with probability $1 - p_t$) chooses the next query by maximizing the sampled functions from all sub-regions (Fig. 1c); see more details in Sec. 3.3.

After initialization, every agent is allowed to query any input in the entire domain $\mathcal{X}$ regardless of the sub-region it is assigned to. So, as $t$ becomes larger, every agent is likely to have explored (and become informative about) more sub-regions in addition to the one it is assigned to. In this regard, for every sub-region $\mathcal{X}_i$, we make the set of weights $\{\varphi_n^{(i)}, \forall n \in [N]\}$ *adaptive* such that they (a) give more weights to those agents exploring $\mathcal{X}_i$ when $t$ is small and (b) gradually become uniform among all agents as $t$ becomes large. We adopt the widely used softmax weighting with temperature $\mathcal{T}$, which has been shown to provide well-calibrated uncertainty for weighted ensemble of GP experts [9]. Concretely, we let $\varphi_n^{(i)} = \frac{\exp((a\mathbb{I}_n^{(i)}+1)/\mathcal{T})}{\sum_{n=1}^N \exp((a\mathbb{I}_n^{(i)}+1)/\mathcal{T})}$,[8] and gradually increase the temperature $\mathcal{T}$ from 1 to $+\infty$ (more details in App. B). The dependence of the weights on $t$ only requires minimal modifications to the algorithm and the theoretical results. So, we drop this dependence for simplicity.

### 3.3 DP-FTS-DE Algorithm

Our complete DP-FTS-DE algorithm after integrating DE (Sec. 3.2) into DP-FTS (Sec. 3.1) is presented in Algo. 1 (central server's role) and Algo. 2 (agent's role), with the steps in circle corresponding to those in Fig. 1a. DP-FTS-DE differs from DP-FTS in two major aspects: Firstly, at initialization ($t = 0$), every agent only explores a local sub-region instead of the entire domain (line 2 of Algo. 2). Secondly, instead of a single vector $\boldsymbol{\omega}_t$, the central server produces and broadcasts $P$ vectors $\{\boldsymbol{\omega}_t^{(i)}, \forall i \in [P]\}$, each corresponding to a different sub-region $\mathcal{X}_i$ and using a different set of weights. Applying different transformations to different vectors (e.g., parameters of different DNN layers) can be naturally incorporated into the general DP framework [45]. Different transformations performed by our central server to produce $P$ vectors can be interpreted as *a single subsampled*

---

[6]For simplicity, we choose the sub-regions to be hyper-rectangles with equal volumes and assign an approximately equal number of agents ($\approx N/P$) to explore every sub-region.

[7]Because its GP surrogate can model the objective function in this local sub-region more accurately [21].

[8]$\mathbb{I}_n^{(i)}$ is an indicator variable that equals 1 if agent $n$ is assigned to explore $\mathcal{X}_i$ and equals 0 otherwise. $a > 0$ is a constant and we set $a = 15$ in all our experiments.

*Gaussian mechanism* producing a single joint vector $\boldsymbol{\omega}_t^{\text{joint}} \triangleq (\boldsymbol{\omega}_t^{(i)})_{i \in [P]}$ (Fig. 1b). Next, we present these transformations performed by the central server (lines 5-12 of Algo. 1) from this perspective.

**Subsampling:** To begin with, after receiving the vectors $\boldsymbol{\omega}_{n,t}$'s from the agents (lines 3-4 of Algo. 1), the central server firstly chooses a random subset of agents $\mathcal{S}_t$ by selecting each agent with probability $q$ (line 6). Next, for every selected agent $n \in \mathcal{S}_t$, the central server constructs a $P \times M$-dimensional joint vector: $\boldsymbol{\omega}_{n,t}^{\text{joint}} \triangleq (N\varphi_n^{(i)}\boldsymbol{\omega}_{n,t})_{i \in [P]}$.

---

**Algorithm 1** DP-FTS-DE (central server)

---

1: $\boldsymbol{\omega}_{-1}^{\text{joint}} = \mathbf{0}$
2: **for** iterations $t = 0, 1, 2, \ldots, T$ **do**
3:     **for** agents $n = 1, 2, \ldots, N$ **in parallel do**
4:         $\boldsymbol{\omega}_{n,t} \leftarrow$ **BO-Agent-$\mathcal{A}_n$**$(t, \boldsymbol{\omega}_{t-1}^{\text{joint}})$     ②
5:     $\boldsymbol{\omega}_t^{(i)} = \mathbf{0}, \forall i \in [P]$
6:     Choose a random subset $\mathcal{S}_t \subset [N]$ of agents
7:     **for** sub-regions $i = 1, 2, \ldots, P$ **do**
8:         **for** agents $n \in \mathcal{S}_t$ **do**
9:             $\widehat{\boldsymbol{\omega}}_{n,t} = \boldsymbol{\omega}_{n,t}\Big/\max\Big(1, \frac{\|\boldsymbol{\omega}_{n,t}\|_2}{S/\sqrt{P}}\Big)$
10:            $\boldsymbol{\omega}_t^{(i)} \mathrel{+}= (\varphi_n^{(i)}/q)\,\widehat{\boldsymbol{\omega}}_{n,t}$
11:     $\boldsymbol{\omega}_t^{(i)} \mathrel{+}= \mathcal{N}\big(\mathbf{0}, (z\varphi_{\max}S/q)^2\mathbf{I}\big)$     ③
12:     Broadcast $\boldsymbol{\omega}_t^{\text{joint}} = (\boldsymbol{\omega}_t^{(i)})_{i \in [P]}$ to all agents     ④
13:     Update the privacy loss using the moments accountant method [4]

---

**Algorithm 2** BO-Agent-$\mathcal{A}_n(t, \boldsymbol{\omega}_{t-1}^{\text{joint}} = (\boldsymbol{\omega}_{t-1}^{(i)})_{i \in [P]})$

---

1: **if** $t = 0$ **then**
2:     Randomly select and query $N_{\text{init}}$ initial points from sub-region $\mathcal{X}_{i_n}$
3: **else**
4:     Sample $r$ from the uniform distribution over $[0, 1]$: $r \sim U(0, 1)$
5:     **if** $r \leq p_t$ **then**
6:         $\mathbf{x}_t^n = \arg\max_{\mathbf{x} \in \mathcal{X}} f_t^n(\mathbf{x})$, where $f_t^n \sim \mathcal{GP}(\mu_t^n(\cdot), \beta_{t+1}^2\sigma_t^n(\cdot, \cdot)^2)$
7:     **else**
8:         $\mathbf{x}_t^n = \arg\max_{\mathbf{x} \in \mathcal{X}} \boldsymbol{\phi}(\mathbf{x})^\top \boldsymbol{\omega}_{t-1}^{(i^{[\mathbf{x}]})}$.[9]
9: Query $\mathbf{x}_t^n$ to observe $y_t^n$
10: Sample $\boldsymbol{\omega}_{n,t}$ and send it to central server     ①, ②

---

**Clipping:** Next, clip every selected vector $\boldsymbol{\omega}_{n,t}$ to obtain $\widehat{\boldsymbol{\omega}}_{n,t}$ such that $\|\widehat{\boldsymbol{\omega}}_{n,t}\|_2 \leq S/\sqrt{P}$. This is equivalent to clipping $\boldsymbol{\omega}_{n,t}^{\text{joint}}$ to obtain: $\widehat{\boldsymbol{\omega}}_{n,t}^{\text{joint}} \triangleq (N\varphi_n^{(i)}\widehat{\boldsymbol{\omega}}_{n,t})_{i \in [P]}$, whose $L_2$ norm is bounded by $\|\widehat{\boldsymbol{\omega}}_{n,t}^{\text{joint}}\|_2 \leq (N^2\varphi_{\max}^2 \sum_{i=1}^{P}\|\widehat{\boldsymbol{\omega}}_{n,t}\|_2^2)^{1/2} \leq N\varphi_{\max}S$ where $\varphi_{\max} \triangleq \max_{i \in [P], n \in [N]} \varphi_n^{(i)}$.

**Weighted Average:** Next, calculate a weighted average of the clipped joint vectors by giving equal weights[10] to all agents: $\boldsymbol{\omega}_t^{\text{joint}} = (qN)^{-1} \sum_{n \in \mathcal{S}_t} \widehat{\boldsymbol{\omega}}_{n,t}^{\text{joint}}$.[11] Note that $\boldsymbol{\omega}_t^{\text{joint}}$ results from the concatenation of the vectors from all sub-regions: $\boldsymbol{\omega}_t^{\text{joint}} = (\boldsymbol{\omega}_t^{(i)})_{i \in [P]}$ where $\boldsymbol{\omega}_t^{(i)} = (qN)^{-1} \sum_{n \in \mathcal{S}_t} N\varphi_n^{(i)}\widehat{\boldsymbol{\omega}}_{n,t} = q^{-1} \sum_{n \in \mathcal{S}_t} \varphi_n^{(i)}\widehat{\boldsymbol{\omega}}_{n,t}$ (line 10 of Algo. 1).

**Gaussian Noise:** Finally, add to each element of $\boldsymbol{\omega}_t^{\text{joint}} = (\boldsymbol{\omega}_t^{(i)})_{i \in [P]}$ a zero-mean Gaussian noise with a standard deviation of $z(N\varphi_{\max}S)/(qN) = z\varphi_{\max}S/q$ (line 11).

Next, the output $\boldsymbol{\omega}_t^{\text{joint}} = (\boldsymbol{\omega}_t^{(i)})_{i \in [P]}$ is broadcast to all agents. After an agent $\mathcal{A}_n$ receives $\boldsymbol{\omega}_t^{\text{joint}}$, with probability $p_{t+1}$, $\mathcal{A}_n$ chooses the next query $\mathbf{x}_{t+1}^n$ by maximizing a function $f_{t+1}^n$ sampled from its own GP posterior $\mathcal{GP}(\mu_t^n(\cdot), \beta_{t+1}^2\sigma_t^n(\cdot, \cdot)^2)$ (1) (line 6 of Algo. 2, $\beta_t \triangleq B + \sigma\sqrt{2(\gamma_{t-1} + 1 + \log(4/\delta)^5)}$); with probability $1 - p_{t+1}$, $\mathcal{A}_n$ uses $\boldsymbol{\omega}_t^{\text{joint}} = [\boldsymbol{\omega}_t^{(i)}]_{i \in [P]}$ received from

---

[9]$i^{[\mathbf{x}]}$ represents the sub-region $\mathbf{x}$ is assigned to.

[10]This weight is only used to aid interpretation and is different from the weights $\varphi_n^{(i)}$'s used in our algorithm.

[11]The summation is divided by $qN$ (i.e., expected number of agents selected) to make it unbiased [47].

the central server to choose $\mathbf{x}_{t+1}^n$ by maximizing the reconstructed functions for all sub-regions (line 8 of Algo. 2), as illustrated in Fig. 1c. Finally, it queries $\mathbf{x}_{t+1}^n$ to observe $y_{t+1}^n$, samples a new vector $\boldsymbol{\omega}_{n,t+1}$ and sends it to the central server (line 10 of Algo. 2), and the algorithm is repeated.

## 4 Theoretical Analysis

In this section, we present theoretical guarantees on both the privacy and utility of our DP-FTS-DE, which combine to yield interesting insights about the *privacy-utility trade-off.*

**Proposition 1** (Privacy Guarantee). *There exist constants $c_1$ and $c_2$ such that for fixed $q$ and $T$ and any $\epsilon < c_1 q^2 T$, $\delta > 0$, DP-FTS-DE (Algo. 1) is $(\epsilon, \delta)$-DP if $z \geq c_2 q\sqrt{T \log(1/\delta)}/\epsilon$.*

Proposition 1 formalizes our privacy guarantee, and its proof (App. A.1) follows directly from Theorem 1 of [1]. Proposition 1 shows that a larger $z$ (i.e., larger variance for Gaussian noise), a smaller $q$ (i.e., smaller expected no. of selected agents) and a smaller $T$ (i.e., smaller no. of iterations) all improve the privacy guarantee because for a fixed $\delta$, they all allow $\epsilon$ to be smaller.

**Theorem 1** (Utility Guarantee). *Assume that all $f^n$'s lie in the RKHS of kernel $k$: $\|f^n\|_k \leq B, \forall n \in [N]$. Let $\gamma_t$ be the max information gain on $f^1$ from any set of $t$ observations, $\varepsilon$ denote an upper bound on the approximation error of RFFs (Sec. 2), $\mathcal{C}_t \triangleq \{n \in [N] \mid \|\boldsymbol{\omega}_{n,t}\|_2 > S/\sqrt{P}\}$, $\delta \in (0, 1)$, $\lambda = 1 + 2/T$ and $\beta_t \triangleq B + \sigma\sqrt{2(\gamma_{t-1} + 1 + \log(4/\delta))}$ (Algo. 2). Choose $(p_t)_{t \in \mathbb{Z}^+}$ to be monotonically increasing s.t. $1 - p_t = \mathcal{O}(1/t^2)$. Then, with probability of at least $1 - \delta$ ($\tilde{\mathcal{O}}$ hides all logarithmic factors),*

$$R_T^1 = \tilde{\mathcal{O}}\Big( \big(B + 1/p_1\big)\gamma_T\sqrt{T} + \textstyle\sum_{t=1}^T \psi_t + B\sum_{t=1}^T \vartheta_t \Big)$$

*where $\psi_t \triangleq \tilde{\mathcal{O}}((1 - p_t)P\varphi_{\max}q^{-1}(\Delta_t + zS\sqrt{M}))$, $\Delta_t \triangleq \sum_{n=1}^N \Delta_{n,t}$, $\Delta_{n,t} \triangleq \tilde{\mathcal{O}}(\varepsilon Bt^2 + B + \sqrt{M} + d_n + \sqrt{\gamma_t})$, and $\vartheta_t \triangleq (1 - p_t)\sum_{i=1}^P \sum_{n \in \mathcal{C}_t} \varphi_n^{(i)}$.*

Theorem 1 (proof in App. A.2) gives an upper bound on the cumulative regret of agent $\mathcal{A}_1$. Note that all three terms in the regret upper bound grow sub-linearly: The first term is sub-linear because $\gamma_T = \mathcal{O}((\log T)^{D+1})$ for the SE kernel ($D$ is the dimension of the input space), and the second and third terms are both sub-linear since we have chosen $1 - p_t = \mathcal{O}(1/t^2)$.[12] So, DP-FTS-DE preserves the no-regret property of FTS [12]. That is, agent $\mathcal{A}_1$ is asymptotically *no-regret* even if all other agents are heterogeneous, i.e., all other agents have significantly different objective functions from $\mathcal{A}_1$. This is particularly desirable because it ensures the robustness of DP-FTS-DE against the heterogeneity of agents, which is an important challenge in both FL and FBO [12, 29] and has also been an important consideration for other works on the theoretical convergence of FL [43, 44]. To prove this robust guarantee, we have upper-bounded the *worst-case error* induced by the use of *any* set of agents, which explains the dependence of the regret bound on $d_n \triangleq \max_{\mathbf{x} \in \mathcal{X}} |f^1(\mathbf{x}) - f^n(\mathbf{x})|$ and $(p_t)_{t \in \mathbb{Z}^+}$. Specifically, larger $d_n$'s indicate larger differences between the objective functions of $\mathcal{A}_1$ and the other agents and hence lead to a worse regret upper bound (through the term $\psi_t$). Smaller values of the sequence $(p_t)_{t \in \mathbb{Z}^+}$ increase the utilization of information from the other agents (line 8 of Algo. 2 becomes more likely to be executed), hence inflating the worst-case error resulting from these information.

Theorem 1, when interpreted together with Proposition 1, also reveals some interesting theoretical insights regarding the **privacy-utility trade-off**. Firstly, a larger $z$ (i.e., larger variance for Gaussian noise) improves the privacy guarantee (Proposition 1) and yet results in a worse utility since it leads to a worse regret upper bound (through $\psi_t$). Secondly, a larger $q$ (i.e., more selected agents in each iteration) improves the utility since it tightens the regret upper bound (by reducing the value of $\psi_t$) at the expense of a worse privacy guarantee (Proposition 1). A general guideline for choosing the values of $z$ and $q$ is to aim for a good utility while ensuring that the privacy loss is within the single-digit range (i.e., $< 10$) [1]. The value of the clipping threshold $S$ exerts no impact on the privacy guarantee. However, $S$ affects the regret upper bound (hence the utility) through two conflicting effects: Firstly, a smaller $S$ reduces the value of $\psi_t$ and hence the regret bound. However, a smaller $S$ is likely to enlarge the cardinality of the set $\mathcal{C}_t$, hence increasing the value of $\vartheta_t$ (Theorem 1). Intuitively, a

---

[12]The requirement of $1 - p_t = \mathcal{O}(1/t^2)$ is needed only to ensure that DP-FTS-DE is no-regret even if all other agents are heterogeneous and is hence a conservative choice. Therefore, it is reasonable to make $1 - p_t$ decrease slower in practice as we show in our experiments (Sec. 5).

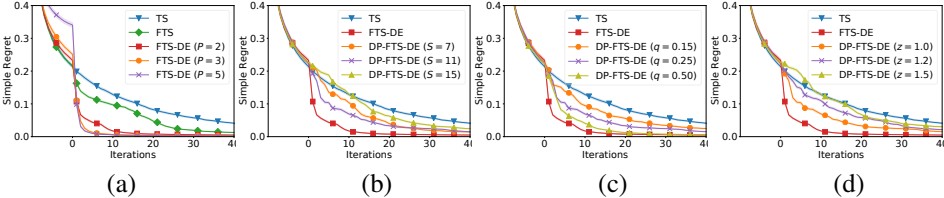

Figure 2: (a) Benefit of DE. (b) Impact of $S$. (c) Impact of $q$; privacy losses after $T = 40$ iterations are $5.93, 9.91, 20.12$ for the respective $q = 0.15, 0.25, 0.5$. (d) Impact of $z$; privacy loss are $9.91, 7.39, 5.22$ for the respective $z = 1.0, 1.2, 1.5$. Every curve is averaged over $N = 200$ agents, each further averaged over 5 runs with different random initializations of size $N_{\text{init}} = 10$. The results before iteration 0 correspond to the initialization period.

smaller $S$ impacts the performance positively by reducing the noise variance (line 11 of Algo. 1) and yet negatively by causing more vectors to be clipped (line 9 of Algo. 1). A general guide on the selection of $S$ is to choose a small value while ensuring that a small number of vectors are clipped.

Regarding the dependence of the regret upper bound on the number of random features $M$, in addition to the dependence through $\Delta_{n,t}$ which has been analyzed by [12], the integration of DP introduces another dependence that implicitly affects the **privacy-utility trade-off**. Besides increasing the value of $\psi_t$, a larger $M$ enlarges the value of $\vartheta_t$ as a larger dimension for the vectors $\boldsymbol{\omega}_{n,t}$'s is expected to increase their $L_2$ norms, hence increasing the cardinality of the set $\mathcal{C}_t$ and consequently the value of $\vartheta_t$ (Theorem 1). So, the additional dependence due to the integration of DP loosens the regret upper bound with an increasing $M$. As a result, if $M$ is larger, we can either *sacrifice privacy to preserve utility* by reducing $z$ or increasing $q$ (both can counter the increase of $\psi_t$), or *sacrifice utility to preserve privacy* by keeping $z$ and $q$ unchanged. The number of sub-regions $P$ also induces a trade-off about the performance of our algorithm, which is partially reflected by Theorem 1. Specifically, the regret bound depends on $P$ through three terms. Two of the terms ($P$ in $\psi_t$ and the summation of $P$ terms in $\vartheta_t$) arise due to the worst-case nature of our regret bound, as discussed earlier: They cause the regret upper bound to increase with $P$ due to the accumulation of the worst-case errors resulting from the $P$ vectors: $\{\boldsymbol{\omega}_t^{(i)}, \forall i \in [P]\}$. Regarding the third term (in the definition of $\mathcal{C}_t$), a larger $P$ is expected to increase the cardinality of the set $\mathcal{C}_t$ (similar to the effect of a larger $M$ discussed above), consequently loosening the regret upper bound by inflating the value of $\vartheta_t$. In this case, as described above, we can choose to sacrifice either privacy or utility. On the other hand, a larger $P$ (i.e., larger number of sub-regions) can improve the practical performance (utility) because it allows every agent to explore only a smaller sub-region which can be better modeled by its GP surrogate (Sec. 3.2). As a result of the worst-case nature of the regret bound mentioned earlier, the latter positive effect leading to better practical utility is not reflected by Theorem 1. Therefore, we instead verify this trade-off induced by $P$ about the practical performance in our experiments (Fig. 6 in App. B.1).

## 5 Experiments

Note that the privacy loss calculated by the moments accountant method is an upper bound on the value of $\epsilon$ for a given value of $\delta$.[4] When calculating the privacy loss, we follow the practice of [47] and set $\delta = 1/N^{1.1}$. All error bars denote standard errors. Due to space constraints, some experimental details are deferred to App. B.

### 5.1 Synthetic Experiments

In synthetic experiments, we firstly sample a function from a GP with the SE kernel, and then apply different small random perturbations to the values of the sampled function to obtain the objective functions of $N = 200$ different agents. We choose $M = 50$ and $1 - p_t = 1/\sqrt{t}, \forall t \in \mathbb{Z}^+$. We firstly demonstrate the performance advantage of modified FTS and FTS-DE (without DP) over standard TS. As shown in Fig. 2a, FTS converges faster than standard TS and more importantly, FTS-DE (Sec. 3.2) further improves the performance of FTS considerably. Moreover, using a larger number $P$ of sub-regions (i.e., smaller sub-regions) brings more performance benefit. This is consistent with our analysis of DE (Sec. 3.2) suggesting that smaller sub-regions are easier to model for the GP surrogate and hence can be better explored. Moreover, we have also verified that after the integration of DP, DP-FTS-DE ($P = 2$) can still achieve significantly better convergence (utility) than DP-FTS for the same level of privacy guarantee (Fig. 4 in App. B.1). These results justify the practical significance of our DE technique (Sec. 3.2). We have also shown (Fig. 5, App. B.1) that both components in our

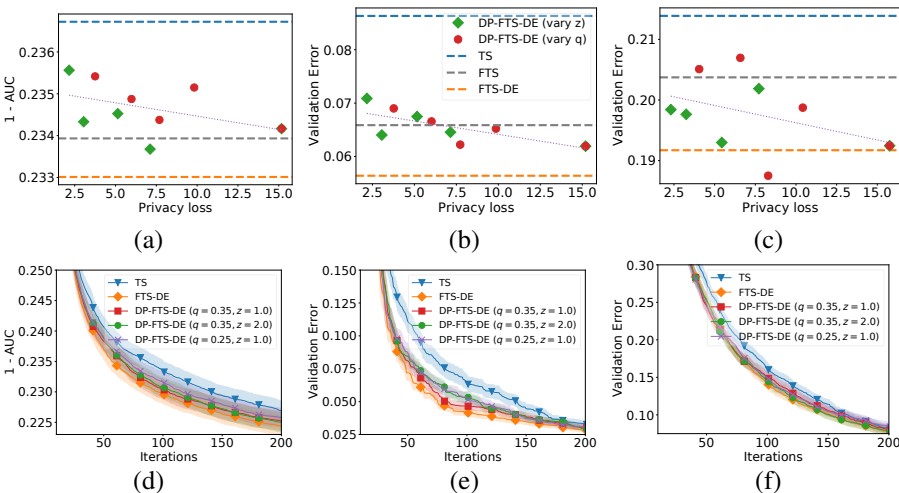

Figure 3: (a,b,c) Privacy loss vs. performance after 60 iterations for landmine detection, human activity recognition, and EMNIST. The more to the *left* (*bottom*), the better the privacy (utility). (d,e,f) Convergence results for some settings in each experiment. Results are averaged over $N$ agents, each further averaged over 100 (a,d) and 10 (b,c,e,f) random initializations ($N_{\text{init}} = 10$). Some error bars overlap because the agents are highly heterogeneous and thus have significantly different scales.

DE technique (i.e., letting every agent explore only a local sub-region and giving more weights to those agents exploring the sub-region) are necessary for the performance of DE.

Fig. 2b explores the impact of the clipping threshold $S$. From Fig. 2b, an overly small $S$ may hinder the performance since it causes too many vectors to be clipped, and an excessively large $S$ may also be detrimental due to increasing the variance of the added Gaussian noise. This corroborates our analysis in Sec. 4. The value of $S = 11$, which delivers the best performance in Fig. 2b, has been chosen such that only a small percentage ($0.8\%$) of the vectors are clipped. Figs. 2c and d show the privacy-utility trade-off of our DP-FTS-DE algorithm induced by $q$ and $z$. The results verify our theoretical insights regarding the impact of the parameters $q$ and $z$ on the privacy-utility trade-off (Sec. 4), i.e., a larger $q$ (smaller $z$) leads to a better utility at the expense of a greater privacy loss. Lastly, we also verify the robustness of our algorithm against agent heterogeneity. In particular, we show that when the objective functions of different agents are significantly different, our FTS-DE algorithm is still able to perform comparably with standard TS and letting the impact of the other agents decay faster (i.e., letting $1 - p_t$ decrease faster) can improve the performance of FTS-DE in this scenario (see Fig. 7 in App. B.1.2).

### 5.2 Real-World Experiments

We adopt 3 commonly used datasets in FL and FBO [12, 60]. We firstly use a landmine detection dataset with $N = 29$ landmine fields [66] and tune 2 hyperparameters of SVM for landmine detection. Next, we use data collected using mobile phone sensors when $N = 30$ subjects are performing 6 activities [2] and tune 3 hyperparameters of logistic regression for activity classification. Lastly, we use the images of handwritten characters by $N = 50$ persons from EMNIST (a commonly used benchmark in FL) [8] and tune 3 hyperparameters of a convolutional neural network used for image classification. In all 3 experiments, we choose $P = 4$, $S = 22.0$, $M = 100$, and $1 - p_t = 1/t$. In the practical deployment of our algorithm, if the values of these parameters are tuned by running additional experiments, the additional privacy loss can be easily accounted for using existing techniques [1].

Figs. 3a,b,c plot the privacy (horizontal axis, more to the left is better) and utility (vertical axis, lower is better) after 60 iterations[13]. The green dots correspond to $z = 1, 1.6, 2, 3, 4$ ($q = 0.35$) and the red dots represent $q = 0.1, 0.15, 0.2, 0.25, 0.35$ ($z = 1$). Results show that with small privacy loss (in the single digit range), DP-FTS-DE achieves a competitive performance (utility) and significantly outperforms standard TS in all settings. The figures also reveal a clear trade-off between privacy and

---

[13]In practice, we recommend that the agents switch to local TS after the value of $1 - p_t$ becomes so small (e.g., after 60 iterations) that the probability of using the information from the central server is negligible (i.e., line 8 of Algo. 2 is unlikely to be executed), after which no privacy loss is incurred.

utility, i.e., a smaller privacy loss (more to the left) generally results in a worse utility (larger vertical value). In addition, these two observations can also be obtained from Figs. 3d,e,f which plot some convergence results: DP-FTS-DE and FTS-DE converge faster than TS; a smaller privacy loss (i.e., larger $z$ or smaller $q$) in general leads to a slower convergence. Figs. 3a,b,c also justify the importance of DE (Sec. 3.2) since FTS-DE (and some settings of DP-FTS-DE) outperforms FTS without DE in all experiments. We also verify the importance of DE when DP is integrated in App. B.2 (Fig. 8). Furthermore, we demonstrate the robustness of our results against the choices of the weights and the number of sub-regions in App. B.2.3. Lastly, our DP-FTS-DE can be easily adapted to use Rényi DP [63][14], which, although requires modifications to our theoretical analysis, offers slightly better privacy loss with comparable performances (Fig. 11 in App. B.2).

## 6 Related Works

BO has been extensively studied recently under different settings [3, 7, 25, 34, 51, 52, 56, 61]. Recent works have added privacy preservation to BO by applying DP to the output of BO [36], using a different notion of privacy other than DP [48], adding DP to outsourced BO [33], or adding local DP to BO [71]. However, none of these works can tackle the FBO setting considered in this paper. Collaborative BO involving multiple agents has also been considered by the work of [59], however, [59] has assumed that all agents share the same objective function and focused on the issue of fairness. Moreover, BO in the multi-agent setting has also been studied from the perspective of game theory [11, 55]. Our method also shares similarity with parallel BO [10, 14, 15, 23, 24, 31]. However, parallel BO optimizes a single objective function while we allow agents to have different objective functions. Our algorithm is also related to multi-fidelity BO [13, 30, 68, 69] because utilizing information received from the central server can be viewed as querying a low-fidelity function. Our DE technique bears similarity to [21] which has also used separate GP surrogates to model different local sub-regions (hyper-rectangles) and shown significantly improved performance.

FL has attracted significant attention in recent years [29, 37, 38, 39, 40, 42, 46]. In particular, privacy preservation using DP has been an important topic for FL [27, 41, 62, 65], including both central DP (with trusted central server) [47] and local [32, 64, 70] or distributed DP [4, 6, 18, 58] (without trusted central server). In addition to our setting of FBO which can be equivalently called *federated GP bandit*, other sequential decision-making problems have also been extended to the federated setting, including federated multi-armed bandit [57, 72], federated linear bandit [17], and federated reinforcement learning [22]. Lastly, federated hyperparameter tuning (i.e., hyperparameter tuning of ML models in the federated setting) has been attracting growing attention recently [26, 35].

## 7 Conclusion and Future Works

We introduce DP-FTS-DE, which equips FBO with rigorous privacy guarantee and is amenable to privacy-utility trade-off both theoretically and empirically. Since our method aims to promote larger-scale adoption of FBO, it may bring the negative societal impact of fairness: some users may become discriminated against by the algorithm. This can be mitigated by extending our method to consider fairness [59], suggesting an interesting future work. A limitation of our work is that the privacy loss offered by the moments accountant method [1] is not state-of-the-art. For example, the more advanced privacy-preserving technique of Gaussian DP [5, 16] can deliver smaller privacy losses than moments accountant and has been widely applied in practice. So, a potential future work is to extend our method to adopt more advanced privacy-preserving techniques such as Gaussian DP. Another limitation of our paper is that we have not accounted for the different network capabilities of different agents, which is a common problem in the federated setting. That is, our algorithm requires every agent to send its updated parameter vector to the central server in every iteration (lines 3 and 4 of Algo. 1), which may not be realistic in some scenarios since the messages from some agents may not be received by the central server. Therefore, accounting for this issue using a principled method represents an interesting future work. Other potential future works include adapting our method for other sequential decision-making problems such as reinforcement learning [22], and incorporating risk aversion [49, 50] into our method for safety-critical domains such as clinical applications (Sec. 1).

---

[14]We only need to modify step 6 of Algo. 1 s.t. we randomly select a fixed number of $Nq$ agents.

## Acknowledgments and Disclosure of Funding

This research/project is supported by the National Research Foundation, Singapore under its Strategic Capability Research Centres Funding Initiative. Any opinions, findings and conclusions or recommendations expressed in this material are those of the author(s) and do not reflect the views of National Research Foundation, Singapore.

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
