# A Proof of Theoretical Results

## A.1 Proof of Proposition 1

Proposition 1 follows directly from the DP guarantee of the works of [1] and [47] (e.g., Theorem 1 of [1]). Therefore, to prove the validity of Proposition 1, we only need to show that the joint subsampled Gaussian mechanism we apply in every iteration (Sec. 3.3) is the same as the one adopted by [1] and [47]. Therefore, we demonstrate here that the interpretation of our privacy-preserving transformations as a single subsampled Gaussian mechanism, which we have described in Sec. 3.3, is equivalent to the transformations adopted by the work of [47].

Firstly, our subsampling step (step 6 of Algo. 1) is the same as the one adopted by [47] since we both use the same subsampling technique, i.e., select every agent with a fixed probability $q$. Secondly, we both clip the (joint) vector from every selected agent (step 9 of Algo. 1) to ensure that its $L_2$ norm is bounded: $\left\|\widehat{\boldsymbol{\omega}}_{n,t}^{\text{joint}}\right\|_2 \leq N\varphi_{\max}S, \forall n \in \mathcal{S}_t$. Thirdly, we have adopted one of the two weighted-average estimators proposed by [47], i.e., the unbiased estimator. Specifically, we set the weight (we follow [47] and denote the weight of agent $\mathcal{A}_n$ by $\omega_n$ here) of every agent to be $\omega_n = 1, \forall n \in [N]$. As a result, the unbiased estimator leads to: $\boldsymbol{\omega}_t^{\text{joint}} = \frac{1}{q\sum_{n=1}^N \omega_n}\sum_{n\in\mathcal{S}_t}\omega_n\widehat{\boldsymbol{\omega}}_{n,t}^{\text{joint}} = \frac{1}{qN}\sum_{n\in\mathcal{S}_t}\widehat{\boldsymbol{\omega}}_{n,t}^{\text{joint}}$. Lastly, we have calculated the sensitivity (which determines the variance of the Gaussian noise) in the same way as [47], i.e., using Lemma 1 of [47]. In particular, note that our clipping step has ensured that $\left\|\omega_n\widehat{\boldsymbol{\omega}}_{n,t}^{\text{joint}}\right\|_2 \leq N\varphi_{\max}S, \forall n \in \mathcal{S}_t$; according to Lemma 1 of [47], we have that the sensitivity can be upper-bounded by: $\mathbb{S} \leq \frac{N\varphi_{\max}S}{q\sum_{n=1}^N \omega_n} = \varphi_{\max}S/q$, which leads to the standard deviation of the Gaussian noise we have added (step 11 of Algo. 1): $z\mathbb{S} = z\varphi_{\max}S/q$.

To conclude, the single joint subsampled Gaussian mechanism performed by our DP-FTS-DE algorithm in every iteration is the same as the one adopted by [47]. Therefore, the DP guarantee of [47] and [1] is also valid for our DP-FTS-DE algorithm, hence justifying the validity of our Proposition 1.

## A.2 Proof of Theorem 1

In this section, we prove Theorem 1, which gives an upper bound on the cumulative regret of agent $\mathcal{A}_1$ running our DP-FTS-DE algorithm. The proof of Theorem 1 makes use of the proof of [12], and the main technical challenge is how to take into account the impacts of (a) our modification to the original FTS algorithm by incorporating a central server and an aggregation through weighted averaging (first paragraph of Sec. 3.1), (b) the privacy-preserving transformations (lines 5-11 of Algo. 1), and (c) distributed exploration (DE) (Sec. 3.2). Since we are mainly interested in the asymptotic regret upper bound, we ignore the impact of the initialization period. Considering initialization would only add a constant term $2BN_{\text{init}}$ to the upper bound on the cumulative regret in Theorem 1 ($N_{\text{init}}$ is the number of initial inputs selected during initialization), and hence would not affect the asymptotic no-regret property of our algorithm.

Note that as we have mentioned in Sections 2 and 4, we prove here an upper bound on the cumulative regret of agent $\mathcal{A}_1$, i.e., $R_T^1 \triangleq \sum_{t=1}^T (f^1(\mathbf{x}^{1,*}) - f^1(\mathbf{x}_t^1))$. To simplify notations, we drop the superscript 1 in the subsequent analysis, i.e., we use $f$ to denote $f^1$, $f_t$ to denote $f_t^1$, $\mathbf{x}_t$ to denote $\mathbf{x}_t^1$, $\mathbf{x}^*$ to denote $\mathbf{x}^{1,*}$, etc. Similarly, we use $\mu_{t-1}$ and $\sigma_{t-1}$ to represent the GP posterior mean and standard deviation of $\mathcal{A}_1$ at iteration $t$.

### A.2.1 Definitions and Supporting Lemmas

We firstly define some notations we use to represent the privacy-preserving transformations, which are consistent with those in the main text. In iteration $t$, we use $\boldsymbol{\omega}_{n,t}$ to denote the vector the central server receives from agent $\mathcal{A}_n$. For a given set of agents $\mathcal{C} \in \{1, \ldots, N\}$, define $\widetilde{\varphi}_{\mathcal{C}}^{(i)} \triangleq \sum_{n\in\mathcal{C}}\varphi_n^{(i)}$, i.e., the total weight of those agents in the set $\mathcal{C}$ for the sub-region $\mathcal{X}_i$. Next, we define $N$ indicator (Bernoulli) random variables $\mathbb{I}_n, \forall n \in [N]$, where $\mathbb{P}(\mathbb{I}_n = 1) = q, \forall n \in [N]$. These indicator random variables will be used to account for the subsampling step (i.e., step 6 of Algo. 1). Denote by $\widehat{\boldsymbol{\omega}}_{n,t}$ the resulting vector after $\boldsymbol{\omega}_{n,t}$ is clipped to have a maximum $L_2$ norm of $S/\sqrt{P}$ (i.e., step 9 of

Algo. 1):

$$\widehat{\boldsymbol{\omega}}_{n,t} \triangleq \frac{\boldsymbol{\omega}_{n,t}}{\max(1, \frac{\|\boldsymbol{\omega}_{n,t}\|_2}{S/\sqrt{P}})}.$$

An immediate consequence is that $\forall \mathbf{x} \in \mathcal{X}$:

$$|\boldsymbol{\phi}(\mathbf{x})^\top \widehat{\boldsymbol{\omega}}_{n,t}| = |\boldsymbol{\phi}(\mathbf{x})^\top \frac{\boldsymbol{\omega}_{n,t}}{\max(1, \frac{\|\boldsymbol{\omega}_{n,t}\|_2}{S/\sqrt{P}})}| = |\boldsymbol{\phi}(\mathbf{x})^\top \boldsymbol{\omega}_{n,t}| \frac{1}{\max(1, \frac{\|\boldsymbol{\omega}_{n,t}\|_2}{S/\sqrt{P}})} \le |\boldsymbol{\phi}(\mathbf{x})^\top \boldsymbol{\omega}_{n,t}|. \quad (3)$$

Denote by $\boldsymbol{\eta}$ the added Gaussian noise vector (i.e., step 11 of Algo. 1): $\boldsymbol{\eta} \sim \mathcal{N}\left(\mathbf{0}, (z\varphi_{\max}S/q)^2\mathbf{I}\right)$. Next, define

$$\boldsymbol{\omega}_t^{(i)} \triangleq \frac{\sum_{n=1}^N \mathbb{I}_n \varphi_n^{(i)} \widehat{\boldsymbol{\omega}}_{n,t}}{q} + \boldsymbol{\eta}. \quad (4)$$

As a result, for agent $\mathcal{A}_1$ at iteration $t > 1$, with probability $1 - p_t$, the query $\mathbf{x}_t^1$ is selected using the $\boldsymbol{\omega}_t^{(i)}$'s: $\mathbf{x}_t^1 = \arg\max_{\mathbf{x} \in \mathcal{X}} \boldsymbol{\phi}(\mathbf{x})^\top \boldsymbol{\omega}_t^{(i^{[\mathbf{x}]})}$, where $i^{[\mathbf{x}]}$ represents the sub-region $\mathbf{x}$ is assigned to. This corresponds to line 8 of Algo. 2. Note that to simplify the notations in the subsequent analyses, we have slightly deviated from the indexing from Algo. 2 by using $\boldsymbol{\omega}_t^{(i^{[\mathbf{x}]})}$ instead of $\boldsymbol{\omega}_{t-1}^{(i^{[\mathbf{x}]})}$. To be consistent with Algo. 2, we can simply replace all appearances of $\boldsymbol{\omega}_t^{(i^{[\mathbf{x}]})}$ by $\boldsymbol{\omega}_{t-1}^{(i^{[\mathbf{x}]})}$ in our proof.

Let $\delta \in (0, 1)$, recall that we have defined in Theorem 1 that $\beta_t \triangleq B + \sigma\sqrt{2(\gamma_{t-1} + 1 + \log(4/\delta)}$ and define $c_t \triangleq \beta_t(1 + \sqrt{2\log(|\mathcal{X}|t^2)})$ for all $t \in \mathbb{Z}^+$. Denote by $A_t$ the event that agent $\mathcal{A}_1$ chooses $\mathbf{x}_t^1$ by maximizing a sampled function from its own GP posterior belief (i.e., $\mathbf{x}_t^1 = \arg\max_{\mathbf{x} \in \mathcal{X}} f_t^1(\mathbf{x})$, as in line 6 of Algo. 2), which happens with probability $p_t$; denote by $B_t$ the event that $\mathcal{A}_1$ chooses $\mathbf{x}_t^1$ using the information received from the central server: $\mathbf{x}_t^1 = \arg\max_{\mathbf{x} \in \mathcal{X}} \boldsymbol{\phi}(\mathbf{x})^\top \boldsymbol{\omega}_t^{(i^{[\mathbf{x}]})}$ (line 8 of Algo. 2), which happens with probability $(1 - p_t)$.

Next, we denote as $\mathcal{F}_t$ the filtration which includes the history of selected inputs and observed outputs of agent $\mathcal{A}_1$ until (including) iteration $t$. Now we define two events that are $\mathcal{F}_{t-1}$-measurable.

**Lemma 1** (Lemma 1 of [12]). *Let $\delta \in (0, 1)$. Define $E^f(t)$ as the event that $|\mu_{t-1}(\mathbf{x}) - f(\mathbf{x})| \le \beta_t\sigma_{t-1}(\mathbf{x})$ for all $\mathbf{x} \in \mathcal{X}$. We have that $\mathbb{P}\left[E^f(t)\right] \ge 1 - \delta/4$ for all $t \ge 1$.*

**Lemma 2** (Lemma 2 of [12]). *Define $E^{f_t}(t)$ as the event that $|f_t(\mathbf{x}) - \mu_{t-1}(\mathbf{x})| \le \beta_t\sqrt{2\log(|\mathcal{X}|t^2)}\sigma_{t-1}(\mathbf{x})$. We have that $\mathbb{P}\left[E^{f_t}(t)|\mathcal{F}_{t-1}\right] \ge 1 - 1/t^2$ for any possible filtration $\mathcal{F}_{t-1}$.*

Note that conditioned on both events $E^f(t)$ and $E^{f_t}(t)$, we have that for all $x \in \mathcal{X}$ and all $t \ge 1$:

$$\begin{aligned}
|f(\mathbf{x}) - f_t(\mathbf{x})| &\le |f(\mathbf{x}) - \mu_{t-1}(\mathbf{x})| + |\mu_{t-1}(\mathbf{x}) - f_t(\mathbf{x})| \\
&= \beta_t\sigma_{t-1}(\mathbf{x}) + \beta_t\sqrt{2\log(|\mathcal{X}|t^2)}\sigma_{t-1}(\mathbf{x}) = c_t\sigma_{t-1}(\mathbf{x}).
\end{aligned} \quad (5)$$

Next, at every iteration $t$, we define a set of *saturated points*, i.e., the set of "bad" inputs at iteration $t$. Intuitively, these inputs are considered as "bad" because their corresponding function values have relatively large differences from the value of the global maximum of $f$.

**Definition 2.** *At iteration $t$, define the set of saturated points as*

$$S_t = \{\mathbf{x} \in \mathcal{X} : \Delta(\mathbf{x}) > c_t\sigma_{t-1}(\mathbf{x})\}$$

*in which $\Delta(\mathbf{x}) \triangleq f(\mathbf{x}^*) - f(\mathbf{x})$ and $\mathbf{x}^* \in \arg\max_{\mathbf{x} \in \mathcal{X}} f(\mathbf{x})$.*

Note that $\Delta(\mathbf{x}^*) \triangleq f(\mathbf{x}^*) - f(\mathbf{x}^*) = 0 < c_t\sigma_{t-1}(\mathbf{x}^*)$ for all $t \ge 1$. Therefore, $\mathbf{x}^*$ is always unsaturated for all $t \ge 1$. $S_t$ is $\mathcal{F}_{t-1}$-measurable.

Consistent with the main text, we define $\widetilde{f}_t^n(\mathbf{x}) \triangleq \boldsymbol{\phi}(\mathbf{x})^\top \boldsymbol{\omega}_{n,t}, \forall \mathbf{x} \in \mathcal{X}$, i.e., $\widetilde{f}_t^n$ is the sampled function from agent $\mathcal{A}_n$'s GP posterior with RFFs approximation at iteration $t$.

**Lemma 3** (Lemma 4 of [12]). *Given any $\delta \in (0, 1)$. We have that for all agents $\mathcal{A}_n, \forall n = 1, \ldots, N$, all $\mathbf{x} \in \mathcal{X}$ and all $t \ge 1$, with probability of at least $1 - \delta/2$,*

$$|\widetilde{f}_t^n(\mathbf{x}) - f^n(\mathbf{x})| \le \tilde{\Delta}_{n,t},$$

*where $\beta'_t = B + \sigma\sqrt{2(\gamma_{t-1} + 1 + \log(8N/\delta))}$, and*

$$\tilde{\Delta}_{n,t} \triangleq \varepsilon\frac{(t+1)^2}{\lambda}\left(B + \sqrt{2\log\left(\frac{4\pi^2 t^2 N}{3\delta}\right)}\right) + \beta'_{t+1} + \sqrt{2\log\frac{2\pi^2 t^2 N}{3\delta}} + M.$$

Note that a difference between our Lemma 3 above and Lemma 4 of [12] is that in their proof, they assumed that the number of observations from agent $\mathcal{A}_n$ is a constant $t_n$; in contrast, we have made use of the fact that in the setting of our DP-FTS-DE algorithm, the number of observations from the other agents are growing with $t$ because all agents are running DP-FTS-DE concurrently. Furthermore, we define

$$\tilde{\Delta}_t^{(i)} \triangleq \sum_{n=1}^{N}\varphi_n^{(i)}\tilde{\Delta}_{n,t}. \tag{6}$$

The next lemma gives a uniform upper bound on the difference between the sampled function $f_t$ from agent $\mathcal{A}_1$ and a weighted combination of the sampled function from all agents, which holds throughout all sub-regions $\mathcal{X}_i, \forall i \in [P]$.

**Lemma 4.** *Denote by $\varepsilon$ an upper bound on the approximation error of RFFs approximation (Sec. 2): $\sup_{\mathbf{x},\mathbf{x}'\in\mathcal{X}}|k(\mathbf{x},\mathbf{x}') - \phi(\mathbf{x})^\top\phi(\mathbf{x}')| \leq \varepsilon$. At iteration $t$, conditioned on the events $E^f(t)$ and $E^{f_t}(t)$, we have that for all $\mathbf{x} \in \mathcal{X}$ and all $i \in [P]$, with probability $\geq 1 - \delta/2$,*

$$|\sum_{n=1}^{N}\varphi_n^{(i)}\widetilde{f}_t^n(\mathbf{x}) - f_t(\mathbf{x})| \leq \Delta_t^{(i)},$$

*in which $\Delta_t^{(i)} \triangleq \sum_{n=1}^{N}\varphi_n^{(i)}\Delta_{n,t}$, and*

$$\begin{aligned}
\Delta_{n,t} &\triangleq \tilde{\Delta}_{n,t} + d_n + c_t \\
&= \varepsilon\frac{(t+1)^2}{\lambda}\left(B + \sqrt{2\log\left(\frac{4\pi^2 t^2 N}{3\delta}\right)}\right) + \beta'_{t+1} + \sqrt{2\log\frac{2\pi^2 t^2 N}{3\delta}} + M + d_n + c_t \quad (7) \\
&= \tilde{\mathcal{O}}(\varepsilon B t^2 + B + \sqrt{M} + d_n + \sqrt{\gamma_t}).
\end{aligned}$$

*Proof.*

$$|\sum_{n=1}^{N}\varphi_n^{(i)}\widetilde{f}_t^n(\mathbf{x}) - f_t(\mathbf{x})| = |\sum_{n=1}^{N}\varphi_n^{(i)}\widetilde{f}_t^n(\mathbf{x}) - \sum_{n=1}^{N}\varphi_n^{(i)}f_t(\mathbf{x})| \leq \sum_{n=1}^{N}\varphi_n^{(i)}|\widetilde{f}_t^n(\mathbf{x}) - f_t(\mathbf{x})|$$

$$\leq \sum_{n=1}^{N}\varphi_n^{(i)}\Delta_{n,t}, \tag{8}$$

where the last inequality results from Lemma 5 of [12]. $\qquad\square$

Note that Lemma 4 above takes into account our modifications to the original FTS algorithm [12] by including a central server and using an aggregation (i.e., weighted average) of the vectors from all agents (first paragraph of Sec. 3.1). Next, define $\Delta_t \triangleq \sum_{n=1}^{N}\Delta_{n,t}$. Note that $\tilde{\Delta}_t^{(i)} \leq \Delta_t^{(i)} \leq \Delta_t, \forall i \in [P]$, and that

$$\sum_{n=1}^{N}\tilde{\Delta}_{n,t} \leq \sum_{n=1}^{N}\Delta_{n,t} = \Delta_t, \tag{9}$$

which will be useful in subsequent proofs.

### A.2.2 Main Proof

The following lemma lower-bounds the probability that the selected input $\mathbf{x}_t$ is unsaturated.

**Lemma 5** (Lemma 7 of [12]). *For any filtration $\mathcal{F}_{t-1}$, conditioned on the event $E^f(t)$, we have that with probability $\geq 1 - \delta/2$,*

$$\mathbb{P}\left(\mathbf{x}_t \in \mathcal{X} \setminus S_t | \mathcal{F}_{t-1}\right) \geq P_t,$$

*in which $P_t \triangleq p_t(p - 1/t^2)$ and $p = \frac{1}{4e\sqrt{\pi}}$.*

*Proof.* Firstly, we have that

$$\mathbb{P}\left(\mathbf{x}_t \in \mathcal{X} \setminus S_t | \mathcal{F}_{t-1}\right) \geq \mathbb{P}\left(\mathbf{x}_t \in \mathcal{X} \setminus S_t | \mathcal{F}_{t-1}, A_t\right)\mathbb{P}(A_t) = \mathbb{P}\left(\mathbf{x}_t \in \mathcal{X} \setminus S_t | \mathcal{F}_{t-1}, A_t\right)p_t. \quad (10)$$

Next, we can lower-bound the probability $\mathbb{P}\left(\mathbf{x}_t \in \mathcal{X} \setminus S_t | \mathcal{F}_{t-1}, A_t\right)$ following Lemma 7 of [12], which leads to $\mathbb{P}\left(\mathbf{x}_t \in \mathcal{X} \setminus S_t | \mathcal{F}_{t-1}, A_t\right) \geq (p - 1/t^2)$ and completes the proof. $\qquad\square$

Next, we derive an upper bound on the expected instantaneous regret of our DP-FTS-DE algorithm.

**Lemma 6.** *For any filtration $\mathcal{F}_{t-1}$, conditioned on the event $E^f(t)$, we have that with probability of $\geq 1 - 5\delta/8$*

$$\mathbb{E}[r_t | \mathcal{F}_{t-1}] \leq c_t\left(1 + \frac{10}{pp_1}\right)\mathbb{E}\left[\sigma_{t-1}(x_t)|\mathcal{F}_{t-1}\right] + 4B\mathbb{E}\left[\vartheta_t | \mathcal{F}_{t-1}\right] + \psi_t + \frac{2B}{t^2},$$

*in which $r_t$ is the instantaneous regret: $r_t \triangleq f(\mathbf{x}^*) - f(\mathbf{x}_t)$, $\vartheta_t \triangleq (1 - p_t)\sum_{i=1}^{P}\widetilde{\varphi}_{\mathcal{C}_t}^{(i)}$, and*

$$\psi_t \triangleq (1 - p_t)P\left[\left(\frac{\varphi_{\max} + 2}{q} + 6\right)\Delta_t + B\left(\frac{2}{q} + \frac{N\varphi_{\max}}{q}\right) + \frac{2zS\varphi_{\max}}{q}\sqrt{2M\log\frac{8M}{\delta}}\right].$$

*Proof.* Firstly, we define $\overline{\mathbf{x}}_t$ as the unsaturated input at iteration $t$ with the smallest posterior standard deviation according to agent $\mathcal{A}_1$'s own GP posterior:

$$\overline{\mathbf{x}}_t \triangleq \arg\min_{\mathbf{x} \in \mathcal{X} \setminus S_t}\sigma_{t-1}(\mathbf{x}). \quad (11)$$

Following this definition, for any $\mathcal{F}_{t-1}$ such that $E^f(t)$ is true, we have that

$$\begin{aligned}
\mathbb{E}\left[\sigma_{t-1}(\mathbf{x}_t)|\mathcal{F}_{t-1}\right] &\geq \mathbb{E}\left[\sigma_{t-1}(\mathbf{x}_t)|\mathcal{F}_{t-1}, \mathbf{x}_t \in \mathcal{X} \setminus S_t\right]\mathbb{P}\left(\mathbf{x}_t \in \mathcal{X} \setminus S_t | \mathcal{F}_{t-1}\right) \\
&\geq \sigma_{t-1}(\overline{\mathbf{x}}_t)P_t,
\end{aligned} \quad (12)$$

in which the last inequality follows from the definition of $\overline{\mathbf{x}}_t$ and Lemma 5.

Next, conditioned on both events $E^f(t)$ and $E^{f_t}(t)$, we have that

$$\begin{aligned}
r_t = \Delta(\mathbf{x}_t) &= f(\mathbf{x}^*) - f(\overline{\mathbf{x}}_t) + f(\overline{\mathbf{x}}_t) - f(\mathbf{x}_t) \\
&\overset{(a)}{\leq} \Delta(\overline{\mathbf{x}}_t) + f_t(\overline{\mathbf{x}}_t) + c_t\sigma_{t-1}(\overline{\mathbf{x}}_t) - f_t(\mathbf{x}_t) + c_t\sigma_{t-1}(\mathbf{x}_t) \\
&\overset{(b)}{\leq} c_t\sigma_{t-1}(\overline{\mathbf{x}}_t) + c_t\sigma_{t-1}(\overline{\mathbf{x}}_t) + c_t\sigma_{t-1}(\mathbf{x}_t) + f_t(\overline{\mathbf{x}}_t) - f_t(\mathbf{x}_t) \\
&= c_t(2\sigma_{t-1}(\overline{\mathbf{x}}_t) + \sigma_{t-1}(\mathbf{x}_t)) + \underline{f_t(\overline{\mathbf{x}}_t) - f_t(\mathbf{x}_t)},
\end{aligned} \quad (13)$$

in which $(a)$ follows from the definition of $\Delta(\mathbf{x})$ and equation (5), and $(b)$ results from the fact that $\overline{\mathbf{x}}_t$ is unsaturated. Denote by $\overline{i}$ the sub-region to which $\overline{\mathbf{x}}_t$ belongs given $\mathcal{F}_{t-1}$. Next, we analyze the

expected value of the underlined term given $\mathcal{F}_{t-1}$:

$$
\mathbb{E}\left[f_t(\overline{\mathbf{x}}_t) - f_t(\mathbf{x}_t)|\mathcal{F}_{t-1}\right]
$$

$$
\stackrel{(a)}{=} \mathbb{P}\left(A_t\right)\mathbb{E}\left[f_t(\overline{\mathbf{x}}_t) - f_t(\mathbf{x}_t)|\mathcal{F}_{t-1}, A_t\right] + \mathbb{P}\left(B_t\right)\mathbb{E}\left[f_t(\overline{\mathbf{x}}_t) - f_t(\mathbf{x}_t)|\mathcal{F}_{t-1}, B_t\right]
$$

$$
\stackrel{(b)}{\leq} \mathbb{P}\left(B_t\right)\mathbb{E}\left[f_t(\overline{\mathbf{x}}_t) - f_t(\mathbf{x}_t)|\mathcal{F}_{t-1}, B_t\right]
$$

$$
\stackrel{(c)}{\leq} \mathbb{P}\left(B_t\right)\sum_{i=1}^{P}\mathbb{P}\left[\mathbf{x}_t \in \mathcal{X}_i\right]\mathbb{E}\left[f_t(\overline{\mathbf{x}}_t) - f_t(\mathbf{x}_t)|\mathcal{F}_{t-1}, B_t, \mathbf{x}_t \in \mathcal{X}_i\right]
$$

$$
\leq \mathbb{P}\left(B_t\right)\sum_{i=1}^{P}\mathbb{E}\left[f_t(\overline{\mathbf{x}}_t) - f_t(\mathbf{x}_t)|\mathcal{F}_{t-1}, B_t, \mathbf{x}_t \in \mathcal{X}_i\right] \tag{14}
$$

$$
\stackrel{(d)}{\leq} \mathbb{P}\left(B_t\right)\sum_{i=1}^{P}\mathbb{E}\left[\sum_{n=1}^{N}\varphi_n^{(i)}\widetilde{f}_t^n(\overline{\mathbf{x}}_t) + \Delta_t^{(i)} - \sum_{n=1}^{N}\varphi_n^{(i)}\widetilde{f}_t^n(\mathbf{x}_t) + \Delta_t^{(i)}\Big|\mathcal{F}_{t-1}, B_t, \mathbf{x}_t \in \mathcal{X}_i\right]
$$

$$
\stackrel{(e)}{\leq} \mathbb{P}\left(B_t\right)\sum_{i=1}^{P}\mathbb{E}\left[\left[\boldsymbol{\phi}(\overline{\mathbf{x}}_t)^\top - \boldsymbol{\phi}(\mathbf{x}_t)^\top\right]\underline{\sum_{n=1}^{N}\varphi_n^{(i)}\boldsymbol{\omega}_{n,t}} + 2\Delta_t^{(i)}\Big|\mathcal{F}_{t-1}, B_t, \mathbf{x}_t \in \mathcal{X}_i\right],
$$

in which $(a)$ and $(c)$ result from the tower rule of expectation; $(b)$ follows since conditioned on the event $A_t$, i.e., $\mathbf{x}_t = \arg\max_{\mathbf{x}\in\mathcal{X}} f_t(\mathbf{x})$, we have that $f_t(\overline{\mathbf{x}}_t) - f_t(\mathbf{x}_t) \leq 0$; $(d)$ results from Lemma 4 and hence holds with probability $\geq 1 - \delta/2$; $(e)$ is a consequence of the definition of $\widetilde{f}_t^n$: $\widetilde{f}_t^n(\mathbf{x}) = \boldsymbol{\phi}(\mathbf{x})^\top\boldsymbol{\omega}_{n,t}, \forall \mathbf{x} \in \mathcal{X}$. Next, we further decompose the underlined term $\sum_{n=1}^{N}\varphi_n^{(i)}\boldsymbol{\omega}_{n,t}$ by:

$$
\sum_{n=1}^{N}\varphi_n^{(i)}\boldsymbol{\omega}_{n,t} = \frac{\sum_{n=1}^{N}q\varphi_n^{(i)}\boldsymbol{\omega}_{n,t}}{q} = \frac{\sum_{n=1}^{N}\mathbb{E}_{\mathbb{I}_n}[\mathbb{I}_n]\varphi_n^{(i)}\boldsymbol{\omega}_{n,t}}{q} = \mathbb{E}_{\mathbb{I}_{1:n}}\left[\frac{\sum_{n=1}^{N}\mathbb{I}_n\varphi_n^{(i)}\boldsymbol{\omega}_{n,t}}{q}\right]
$$

$$
= \mathbb{E}_{\mathbb{I}_{1:n}}\left[\frac{\sum_{n=1}^{N}\mathbb{I}_n\varphi_n^{(i)}\boldsymbol{\omega}_{n,t}}{q} - \frac{\sum_{n=1}^{N}\mathbb{I}_n\varphi_n^{(i)}\widehat{\boldsymbol{\omega}}_{n,t}}{q} - \boldsymbol{\eta} + \frac{\sum_{n=1}^{N}\mathbb{I}_n\varphi_n^{(i)}\widehat{\boldsymbol{\omega}}_{n,t}}{q} + \boldsymbol{\eta}\right] \tag{15}
$$

$$
= \mathbb{E}_{\mathbb{I}_{1:n}}\left[\frac{\sum_{n=1}^{N}\mathbb{I}_n\varphi_n^{(i)}\boldsymbol{\omega}_{n,t}}{q} - \frac{\sum_{n=1}^{N}\mathbb{I}_n\varphi_n^{(i)}\widehat{\boldsymbol{\omega}}_{n,t}}{q} - \boldsymbol{\eta} + \boldsymbol{\omega}_t^{(i)}\right],
$$

where in the last equality we have made use of the definition of $\boldsymbol{\omega}_t^{(i)}$ (4). Next, we plug (15) back into (14):

$$
\mathbb{E}\left[f_t(\overline{\mathbf{x}}_t) - f_t(\mathbf{x}_t)|\mathcal{F}_{t-1}\right]
$$

$$
\leq \mathbb{P}\left(B_t\right)\sum_{i=1}^{P}\mathbb{E}\left[\left[\boldsymbol{\phi}(\overline{\mathbf{x}}_t)^\top - \boldsymbol{\phi}(\mathbf{x}_t)^\top\right]\mathbb{E}_{\mathbb{I}_{1:n}}\left[\frac{\sum_{n=1}^{N}\mathbb{I}_n\varphi_n^{(i)}\boldsymbol{\omega}_{n,t}}{q} - \frac{\sum_{n=1}^{N}\mathbb{I}_n\varphi_n^{(i)}\widehat{\boldsymbol{\omega}}_{n,t}}{q} - \boldsymbol{\eta}\right] + \right.
$$

$$
\left.\left[\boldsymbol{\phi}(\overline{\mathbf{x}}_t)^\top\mathbb{E}_{\mathbb{I}_{1:n}}[\boldsymbol{\omega}_t^{(i)}] - \boldsymbol{\phi}(\mathbf{x}_t)^\top\mathbb{E}_{\mathbb{I}_{1:n}}[\boldsymbol{\omega}_t^{(i)}]\right] + 2\Delta_t^{(i)}\Big|\mathcal{F}_{t-1}, B_t, \mathbf{x}_t \in \mathcal{X}_i\right]
$$

$$
\leq \mathbb{P}\left(B_t\right)\sum_{i=1}^{P}\mathbb{E}\left[\underbrace{\left[\boldsymbol{\phi}(\overline{\mathbf{x}}_t)^\top - \boldsymbol{\phi}(\mathbf{x}_t)^\top\right]\mathbb{E}_{\mathbb{I}_{1:n}}\left[\frac{\sum_{n=1}^{N}\mathbb{I}_n\varphi_n^{(i)}\boldsymbol{\omega}_{n,t}}{q} - \frac{\sum_{n=1}^{N}\mathbb{I}_n\varphi_n^{(i)}\widehat{\boldsymbol{\omega}}_{n,t}}{q}\right]}_{A_1} + \right.
$$

$$
\left[\underbrace{\boldsymbol{\phi}(\overline{\mathbf{x}}_t)^\top\mathbb{E}_{\mathbb{I}_{1:n}}[\boldsymbol{\omega}_t^{(i)}] - \boldsymbol{\phi}(\overline{\mathbf{x}}_t)^\top\boldsymbol{\omega}_t^{(i)}}_{A_2} + \underbrace{\boldsymbol{\phi}(\overline{\mathbf{x}}_t)^\top\boldsymbol{\omega}_t^{(i)} - \boldsymbol{\phi}(\overline{\mathbf{x}}_t)^\top\boldsymbol{\omega}_t^{(\bar{i})}}_{A_3} + \underbrace{\boldsymbol{\phi}(\overline{\mathbf{x}}_t)^\top\boldsymbol{\omega}_t^{(\bar{i})} - \boldsymbol{\phi}(\mathbf{x}_t)^\top\boldsymbol{\omega}_t^{(i)}}_{A_4} + \right.
$$

$$
\left.\underbrace{\boldsymbol{\phi}(\mathbf{x}_t)^\top\boldsymbol{\omega}_t^{(i)} - \boldsymbol{\phi}(\mathbf{x}_t)^\top\mathbb{E}_{\mathbb{I}_{1:n}}[\boldsymbol{\omega}_t^{(i)}]}_{A_5}\right] - \underbrace{\left[\boldsymbol{\phi}(\overline{\mathbf{x}}_t)^\top - \boldsymbol{\phi}(\mathbf{x}_t)^\top\right]\boldsymbol{\eta}}_{A_6} + 2\Delta_t^{(i)}\Big|\mathcal{F}_{t-1}, B_t, \mathbf{x}_t \in \mathcal{X}_i\right]
$$

$$
\tag{16}
$$

Next, we separately upper-bound the terms $A_1$ to $A_6$. Firstly, we bound the term $A_1$. Define $\mathcal{C}_t \triangleq \{n \in [N] \big| \|\boldsymbol{\omega}_{n,t}\|_2 > S/\sqrt{P}\}$, which is the same as the definition in Theorem 1. That is, $\mathcal{C}_t$ contains the indices of those agents whose vector of $\boldsymbol{\omega}_{n,t}$ has a larger $L_2$ norm than $S/\sqrt{P}$ in iteration $t$. $A_1$ can thus be analyzed as:

$$
\begin{aligned}
&\left| \left[ \boldsymbol{\phi}(\overline{\mathbf{x}}_t)^\top - \boldsymbol{\phi}(\mathbf{x}_t)^\top \right] \mathbb{E}_{\mathbb{I}_{1:n}} \left[ \frac{\sum_{n=1}^{N} \mathbb{I}_n \varphi_n^{(i)} \boldsymbol{\omega}_{n,t}}{q} - \frac{\sum_{n=1}^{N} \mathbb{I}_n \varphi_n^{(i)} \widehat{\boldsymbol{\omega}}_{n,t}}{q} \right] \right| \\
&\overset{(a)}{=} \left| \left[ \boldsymbol{\phi}(\overline{\mathbf{x}}_t)^\top - \boldsymbol{\phi}(\mathbf{x}_t)^\top \right] \sum_{n=1}^{N} \varphi_n^{(i)} (\boldsymbol{\omega}_{n,t} - \widehat{\boldsymbol{\omega}}_{n,t}) \right| \\
&\overset{(b)}{=} \left| \left[ \boldsymbol{\phi}(\overline{\mathbf{x}}_t)^\top - \boldsymbol{\phi}(\mathbf{x}_t)^\top \right] \sum_{n \in \mathcal{C}_t} \varphi_n^{(i)} (\boldsymbol{\omega}_{n,t} - \widehat{\boldsymbol{\omega}}_{n,t}) \right| \\
&= \left| \sum_{n \in \mathcal{C}_t} \varphi_n^{(i)} \left[ \boldsymbol{\phi}(\overline{\mathbf{x}}_t)^\top - \boldsymbol{\phi}(\mathbf{x}_t)^\top \right] \left[ \boldsymbol{\omega}_{n,t} - \widehat{\boldsymbol{\omega}}_{n,t} \right] \right| \\
&= \left| \sum_{n \in \mathcal{C}_t} \varphi_n^{(i)} \left[ \boldsymbol{\phi}(\overline{\mathbf{x}}_t)^\top \boldsymbol{\omega}_{n,t} + \boldsymbol{\phi}(\mathbf{x}_t)^\top \widehat{\boldsymbol{\omega}}_{n,t} - \boldsymbol{\phi}(\overline{\mathbf{x}}_t)^\top \widehat{\boldsymbol{\omega}}_{n,t} - \boldsymbol{\phi}(\mathbf{x}_t)^\top \boldsymbol{\omega}_{n,t} \right] \right| \\
&\leq \sum_{n \in \mathcal{C}_t} \varphi_n^{(i)} \left[ \left| \boldsymbol{\phi}(\overline{\mathbf{x}}_t)^\top \boldsymbol{\omega}_{n,t} \right| + \left| \boldsymbol{\phi}(\mathbf{x}_t)^\top \widehat{\boldsymbol{\omega}}_{n,t} \right| + \left| \boldsymbol{\phi}(\overline{\mathbf{x}}_t)^\top \widehat{\boldsymbol{\omega}}_{n,t} \right| + \left| \boldsymbol{\phi}(\mathbf{x}_t)^\top \boldsymbol{\omega}_{n,t} \right| \right] \\
&\leq \sum_{n \in \mathcal{C}_t} \varphi_n^{(i)} \left[ \left| \boldsymbol{\phi}(\overline{\mathbf{x}}_t)^\top \boldsymbol{\omega}_{n,t} \right| + \left| \boldsymbol{\phi}(\mathbf{x}_t)^\top \boldsymbol{\omega}_{n,t} \right| + \left| \boldsymbol{\phi}(\overline{\mathbf{x}}_t)^\top \boldsymbol{\omega}_{n,t} \right| + \left| \boldsymbol{\phi}(\mathbf{x}_t)^\top \boldsymbol{\omega}_{n,t} \right| \right] \\
&\leq 2 \sum_{n \in \mathcal{C}_t} \varphi_n^{(i)} \left[ \left| \widetilde{f}_t^n(\overline{\mathbf{x}}_t) \right| + \left| \widetilde{f}_t^n(\mathbf{x}) \right| \right] \\
&\overset{(c)}{\leq} 2 \sum_{n \in \mathcal{C}_t} \varphi_n^{(i)} (\tilde{\Delta}_{n,t} + B + \tilde{\Delta}_{n,t} + B) \\
&= 4 \sum_{n \in \mathcal{C}_t} \varphi_n^{(i)} \tilde{\Delta}_{n,t} + 4 \sum_{n \in \mathcal{C}_t} \varphi_n^{(i)} B \\
&\overset{(d)}{\leq} 4 \sum_{n=1}^{N} \varphi_n^{(i)} \tilde{\Delta}_{n,t} + 4B \widetilde{\varphi}_{\mathcal{C}_t}^{(i)} \\
&\overset{(e)}{=} 4 \left( \tilde{\Delta}_t^{(i)} + B \widetilde{\varphi}_{\mathcal{C}_t}^{(i)} \right),
\end{aligned}
\tag{17}
$$

in which $(a)$ follows since $\mathbb{E}_n[\mathbb{I}_n] = q, \forall n \in [N]$; $(b)$ follows since for those agents $n \notin \mathcal{C}_t$, $\boldsymbol{\omega}_{n,t} - \widehat{\boldsymbol{\omega}}_{n,t} = \mathbf{0}$ because the vector $\boldsymbol{\omega}_{n,t}$ is not clipped; $(c)$ results from Lemma 3 and that $|f^n(\mathbf{x})| \leq B, \forall \mathbf{x} \in \mathcal{X}, n \in [N]$ (this is because of our assumption that $\|f^n\|_k \leq B, \forall n \in [N]$, Sec. 2); $(d)$ follows from the definition of $\widetilde{\varphi}_{\mathcal{C}_t}^{(i)} \triangleq \sum_{n \in \mathcal{C}_t} \varphi_n^{(i)}$; $(e)$ results from the definition of $\tilde{\Delta}_t^{(i)}$ (6).

Subsequently, we upper-bound the terms $A_2$ and $A_5$. For any $\mathbf{x} \in \mathcal{X}$, we have that

$$
\left| \phi(\mathbf{x})^\top \mathbb{E}_{\mathbb{I}_{1:n}}[\boldsymbol{\omega}_t^{(i)}] - \phi(\mathbf{x})^\top \boldsymbol{\omega}_t^{(i)} \right| = \left| \phi(\mathbf{x})^\top \left( \mathbb{E}_{\mathbb{I}_{1:n}} \left[ \frac{\sum_{n=1}^N \mathbb{I}_n \varphi_n^{(i)} \widehat{\boldsymbol{\omega}}_{n,t}}{q} \right] - \frac{\sum_{n=1}^N \mathbb{I}_n \varphi_n^{(i)} \widehat{\boldsymbol{\omega}}_{n,t}}{q} \right) \right|
$$

$$
= \left| \phi(\mathbf{x})^\top \left( \frac{\sum_{n=1}^N q \varphi_n^{(i)} \widehat{\boldsymbol{\omega}}_{n,t}}{q} - \frac{\sum_{n=1}^N \mathbb{I}_n \varphi_n^{(i)} \widehat{\boldsymbol{\omega}}_{n,t}}{q} \right) \right|
$$

$$
= \left| \phi(\mathbf{x})^\top \frac{1}{q} \sum_{n=1}^N (q - \mathbb{I}_n) \varphi_n^{(i)} \widehat{\boldsymbol{\omega}}_{n,t} \right| \leq \frac{1}{q} \sum_{n=1}^N \left| (q - \mathbb{I}_n) \varphi_n^{(i)} \phi(\mathbf{x})^\top \widehat{\boldsymbol{\omega}}_{n,t} \right|
$$

$$
\overset{(a)}{\leq} \frac{1}{q} \sum_{n=1}^N \varphi_n^{(i)} \left| \phi(\mathbf{x})^\top \widehat{\boldsymbol{\omega}}_{n,t} \right| \overset{(b)}{\leq} \frac{1}{q} \sum_{n=1}^N \varphi_n^{(i)} \left| \phi(\mathbf{x})^\top \boldsymbol{\omega}_{n,t} \right|
$$

$$
= \frac{1}{q} \sum_{n=1}^N \varphi_n^{(i)} \left| \widetilde{f}_t^n(\mathbf{x}) \right| = \frac{1}{q} \sum_{n=1}^N \varphi_n^{(i)} \left| \widetilde{f}_t^n(\mathbf{x}) - f^n(\mathbf{x}) + f^n(\mathbf{x}) \right|
$$

$$
\leq \frac{1}{q} \sum_{n=1}^N \varphi_n^{(i)} \left( \left| \widetilde{f}_t^n(\mathbf{x}) - f^n(\mathbf{x}) \right| + \left| f^n(\mathbf{x}) \right| \right)
$$

$$
\overset{(c)}{\leq} \frac{1}{q} \sum_{n=1}^N \varphi_n^{(i)} \left( \widetilde{\Delta}_{n,t} + B \right)
$$

$$
\overset{(d)}{=} \frac{1}{q} \left( \widetilde{\Delta}_t^{(i)} + B \right),
$$

$$(18)$$

in which $(a)$ follows since $|q - \mathbb{I}_n| \leq 1$; $(b)$ results from (3); $(c)$ results from Lemma 3 and that $|f^n(\mathbf{x})| \leq B, \forall \mathbf{x} \in \mathcal{X}, n \in [N]$; $(d)$ results from the definition of $\widetilde{\Delta}_t^{(i)}$ (6).

Next, we upper-bound the term $A_3$, which arises because the sub-regions $i$ and $\bar{i}$ may be different. We have that for any $\mathbf{x} \in \mathcal{X}$,

$$
\left| \phi(\mathbf{x})^\top \boldsymbol{\omega}_t^{(i)} - \phi(\mathbf{x})^\top \boldsymbol{\omega}_t^{(\bar{i})} \right| = \left| \phi(\mathbf{x})^\top \left( \boldsymbol{\omega}_t^{(i)} - \boldsymbol{\omega}_t^{(\bar{i})} \right) \right|
$$

$$
= \left| \phi(\mathbf{x})^\top \left( \frac{\sum_{n=1}^N \mathbb{I}_n \varphi_n^{(i)} \widehat{\boldsymbol{\omega}}_{n,t}}{q} - \frac{\sum_{n=1}^N \mathbb{I}_n \varphi_n^{(\bar{i})} \widehat{\boldsymbol{\omega}}_{n,t}}{q} \right) \right|
$$

$$
\leq \frac{1}{q} \sum_{n=1}^N \mathbb{I}_n \left| \varphi_n^{(i)} - \varphi_n^{(\bar{i})} \right| \left| \phi(\mathbf{x})^\top \widehat{\boldsymbol{\omega}}_{n,t} \right|
$$

$$
\leq \frac{1}{q} \sum_{n=1}^N \varphi_{\max} \left| \phi(\mathbf{x})^\top \widehat{\boldsymbol{\omega}}_{n,t} \right| \qquad (19)
$$

$$
\overset{(a)}{\leq} \frac{1}{q} \sum_{n=1}^N \varphi_{\max} \left| \phi(\mathbf{x})^\top \boldsymbol{\omega}_{n,t} \right|
$$

$$
\overset{(b)}{\leq} \frac{1}{q} \sum_{n=1}^N \varphi_{\max} \left( \widetilde{\Delta}_{n,t} + B \right)
$$

$$
\overset{(c)}{\leq} \frac{\varphi_{\max}}{q} \left( \Delta_t + NB \right),
$$

$(a)$ follows because of (3); $(b)$ results from Lemma 3 and that $|f^n(\mathbf{x})| \leq B, \forall \mathbf{x} \in \mathcal{X}, n \in [N]$; $(c)$ follows from (9).

Next, regarding $A_4$, note that conditioned on the event $B_t$, $\mathbf{x}_t$ is selected by: $\mathbf{x}_t = \arg\max_{\mathbf{x} \in \mathcal{X}} \phi(\mathbf{x})^\top \boldsymbol{\omega}_t^{(i^{[\mathbf{x}]})}$ in which $i^{[\mathbf{x}]}$ represents the sub-region $\mathbf{x}$ belongs to. Therefore, be-

cause $\overline{\mathbf{x}}_t \in \mathcal{X}_{\overline{i}}$ and $\mathbf{x}_t \in \mathcal{X}_i$ (since we are conditioning on this event), we have that $\phi(\overline{\mathbf{x}}_t)^\top \boldsymbol{\omega}_t^{(\overline{i})} - \phi(\mathbf{x}_t)^\top \boldsymbol{\omega}_t^{(i)} \leq 0$. In other words, $A_4 \leq 0$.

Finally, The term $A_6$ can be upper-bounded using standard Gaussian concentration inequality:

$$
\begin{aligned}
\left| \left[ \phi(\overline{\mathbf{x}}_t)^\top - \phi(\mathbf{x}_t)^\top \right] \boldsymbol{\eta} \right| &\leq \left\| \phi(\overline{\mathbf{x}}_t) - \phi(\mathbf{x}_t) \right\|_2 \|\boldsymbol{\eta}\|_2 \\
&\leq \left( \left\| \phi(\overline{\mathbf{x}}_t) \right\|_2 + \left\| \phi(\mathbf{x}_t) \right\|_2 \right) \|\boldsymbol{\eta}\|_2 \overset{(a)}{\leq} 2\|\boldsymbol{\eta}\|_2 \\
&\overset{(b)}{\leq} \frac{2zS\varphi_{\max}}{q} \sqrt{2M \log \frac{8M}{\delta}},
\end{aligned}
\tag{20}
$$

where $(a)$ follows since the random features have been constructed such that $\left\| \phi(\mathbf{x}) \right\|_2^2 = \sigma_0^2 \leq 1$ [12], and $(b)$ follows from standard Gaussian concentration inequality and hence holds with probability $> 1 - \delta/8$.

Now we can exploit the upper bounds on the terms $A_1$ to $A_6$ we have derived above (equations (17), (18), (19), (20)), and continue to upper-bound $\mathbb{E}\left[ f_t(\overline{\mathbf{x}}_t) - f_t(\mathbf{x}_t) | \mathcal{F}_{t-1} \right]$ following (16):

$$
\begin{aligned}
\mathbb{E}[f_t(\overline{\mathbf{x}}_t) - f_t(\mathbf{x}_t)|\mathcal{F}_{t-1}] &\leq \mathbb{P}(B_t) \sum_{i=1}^{P} \mathbb{E}\Bigg[ \underbrace{4\left( \tilde{\Delta}_t^{(i)} + B\widetilde{\varphi}_{\mathcal{C}_t}^{(i)} \right)}_{A_1} + \underbrace{\frac{2}{q}\left( \tilde{\Delta}_t^{(i)} + B \right)}_{A_2 + A_5} + \\
&\quad \underbrace{\frac{\varphi_{\max}}{q}\left( \Delta_t + NB \right)}_{A_3} + \underbrace{\frac{2zS\varphi_{\max}}{q}\sqrt{2M\log\frac{8M}{\delta}}}_{A_6} + 2\Delta_t^{(i)} \Bigg| \mathcal{F}_{t-1}, B_t, \mathbf{x}_t \in \mathcal{X}_i \Bigg] \\
&= (1 - p_t) \sum_{i=1}^{P} \Bigg[ 4\left( \tilde{\Delta}_t^{(i)} + B\mathbb{E}\left[ \widetilde{\varphi}_{\mathcal{C}_t}^{(i)} | \mathcal{F}_{t-1} \right] \right) + \frac{2}{q}\left( \tilde{\Delta}_t^{(i)} + B \right) + \\
&\quad \frac{\varphi_{\max}}{q}\left( \Delta_t + NB \right) + \frac{2zS\varphi_{\max}}{q}\sqrt{2M\log\frac{8M}{\delta}} + 2\Delta_t^{(i)} \Bigg] \\
&= (1 - p_t) \sum_{i=1}^{P} \Bigg[ 4B\mathbb{E}\left[ \widetilde{\varphi}_{\mathcal{C}_t}^{(i)} | \mathcal{F}_{t-1} \right] + \left( \frac{2}{q} + 4 \right)\tilde{\Delta}_t^{(i)} + 2\Delta_t^{(i)} + \frac{\varphi_{\max}}{q}\Delta_t + \\
&\quad B\left( \frac{2}{q} + \frac{N\varphi_{\max}}{q} \right) + \frac{2zS\varphi_{\max}}{q}\sqrt{2M\log\frac{8M}{\delta}} \Bigg] \\
&\overset{(a)}{\leq} (1 - p_t) \sum_{i=1}^{P} \Bigg[ 4B\mathbb{E}\left[ \widetilde{\varphi}_{\mathcal{C}_t}^{(i)} | \mathcal{F}_{t-1} \right] + \left( \frac{2}{q} + 6 + \frac{\varphi_{\max}}{q} \right)\Delta_t + \\
&\quad B\left( \frac{2}{q} + \frac{N\varphi_{\max}}{q} \right) + \frac{2zS\varphi_{\max}}{q}\sqrt{2M\log\frac{8M}{\delta}} \Bigg] \\
&= 4B\mathbb{E}\left[ (1 - p_t)\sum_{i=1}^{P} \widetilde{\varphi}_{\mathcal{C}_t}^{(i)} | \mathcal{F}_{t-1} \right] + (1 - p_t)\Bigg[ P\left( \frac{2}{q} + 6 + \frac{\varphi_{\max}}{q} \right)\Delta_t + \\
&\quad PB\left( \frac{2}{q} + \frac{N\varphi_{\max}}{q} \right) + P\frac{2zS\varphi_{\max}}{q}\sqrt{2M\log\frac{8M}{\delta}} \Bigg] \\
&= 4B\mathbb{E}\left[ \vartheta_t | \mathcal{F}_{t-1} \right] + \psi_t,
\end{aligned}
\tag{21}
$$

where $(a)$ follows because $\tilde{\Delta}_t^{(i)} \leq \Delta_t^{(i)} \leq \Delta_t, \forall i \in [P]$. In the last equality, we have made use of the definitions of $\vartheta_t$ and $\psi_t$. Note that since we have made use of Lemma 4 (14) which holds with probability $\geq 1 - \delta/2$, and Gaussian concentration inequality (20) which holds with probability $\geq 1 - \delta/8$, equation (21) holds with probability $\geq 1 - \delta/2 - \delta/8 = 1 - 5\delta/8$.

Finally, we plug (21) back into (13):

$$
\mathbb{E}\left[r_t | \mathcal{F}_{t-1}\right]
$$

$$
\leq \mathbb{E}\left[c_t(2\sigma_{t-1}(\overline{\mathbf{x}}_t) + \sigma_{t-1}(\mathbf{x}_t)) + f_t(\overline{\mathbf{x}}_t) - f_t(\mathbf{x}_t)|\mathcal{F}_{t-1}\right] + 2B\mathbb{P}\left[\overline{E^{f_t}(t)}|\mathcal{F}_{t-1}\right]
$$

$$
\leq \mathbb{E}\left[c_t(2\sigma_{t-1}(\overline{\mathbf{x}}_t) + \sigma_{t-1}(\mathbf{x}_t))|\mathcal{F}_{t-1}\right] + \mathbb{E}\left[f_t(\overline{\mathbf{x}}_t) - f_t(\mathbf{x}_t)|\mathcal{F}_{t-1}\right] + 2B\mathbb{P}\left[\overline{E^{f_t}(t)}|\mathcal{F}_{t-1}\right]
$$

$$
\overset{(a)}{\leq} \frac{2c_t}{P_t}\mathbb{E}\left[\sigma_{t-1}(\mathbf{x}_t)|\mathcal{F}_{t-1}\right] + c_t\mathbb{E}\left[\sigma_{t-1}(\mathbf{x}_t)|\mathcal{F}_{t-1}\right] + 4B\mathbb{E}\left[\vartheta_t|\mathcal{F}_{t-1}\right] + \psi_t + \frac{2B}{t^2}
$$

$$
\leq c_t\left(1 + \frac{2}{P_t}\right)\mathbb{E}\left[\sigma_{t-1}(\mathbf{x}_t)|\mathcal{F}_{t-1}\right] + 4B\mathbb{E}\left[\vartheta_t|\mathcal{F}_{t-1}\right] + \psi_t + \frac{2B}{t^2}
$$

$$
\overset{(b)}{\leq} c_t\left(1 + \frac{10}{pp_1}\right)\mathbb{E}\left[\sigma_{t-1}(\mathbf{x}_t)|\mathcal{F}_{t-1}\right] + 4B\mathbb{E}\left[\vartheta_t|\mathcal{F}_{t-1}\right] + \psi_t + \frac{2B}{t^2},
$$

$$(22)$$

in which $(a)$ follows from (12) and (21), and $(b)$ follows since:

$$
\frac{2}{P_t} = \frac{2}{p_t(p - \frac{1}{t^2})} \leq \frac{10}{pp_t} \leq \frac{10}{pp_1}, \tag{23}
$$

which was valid because $1/(p - 1/t^2) \leq 5/p$ and $p_t \geq p_1$ for all $t \geq 1$.

Note that since the proof of (22) makes use of (21), therefore, (22), as well as Lemma 6, also holds with probability of $\geq 1 - 5\delta/8$.

$\square$

**Lemma 7.** *Given $\delta \in (0,1)$, then with probability of at least $1 - \delta$,*

$$
R_T \leq c_T\left(1 + \frac{10}{pp_1}\right)\mathcal{O}(\sqrt{T\gamma_T}) + \sum_{t=1}^{T}\psi_t + \frac{B\pi^2}{3} + 4B\sum_{t=1}^{T}\vartheta_t +
$$

$$
\left[c_T\left(1 + \frac{4B}{p} + \frac{10}{pp_1}\right) + \psi_1 + 4B\right]\sqrt{2T\log\frac{8}{\delta}},
$$

*in which $\gamma_T$ is the maximum information gain about $f$ obtained from any set of $T$ observations.*

*Proof.* The proof resembles the that of Lemma 11 of [12], and is hence omitted. A difference from Lemma 11 of [12] is that an error probability of $\delta/8$ has been used in the Azuma-Hoeffding Inequality in the proof here. $\square$

Finally, we are ready to prove Theorem 1. Recall that $c_t = \mathcal{O}\left(\left(B + \sqrt{\gamma_t + \log(1/\delta)}\right)\sqrt{\log t}\right)$. Therefore,

$$
R_T = \mathcal{O}\left(\frac{1}{p_1}\left(B + \sqrt{\gamma_T + \log\frac{1}{\delta}}\right)\sqrt{\log T}\sqrt{T\gamma_T} + \sum_{t=1}^{T}\psi_t + B\sum_{t=1}^{T}\vartheta_t + \right.
$$

$$
\left.\left(B + \frac{1}{p_1}\right)\left(B + \sqrt{\gamma_T + \log\frac{1}{\delta}}\right)\sqrt{\log T}\sqrt{T\log\frac{1}{\delta}}\right)
$$

$$
= \mathcal{O}\left(\left(B + \frac{1}{p_1}\right)\sqrt{T\log T\gamma_T\log\frac{1}{\delta}\left(\gamma_T + \log\frac{1}{\delta}\right)} + \sum_{t=1}^{T}\psi_t + B\sum_{t=1}^{T}\vartheta_t\right)
$$

$$(24)$$

$$
= \tilde{\mathcal{O}}\left(\left(B + \frac{1}{p_1}\right)\gamma_T\sqrt{T} + \sum_{t=1}^{T}\psi_t + B\sum_{t=1}^{T}\vartheta_t\right),
$$

which finally completes the proof.

# B  Experiments

As we have mentioned in the main text (last paragraph of Sec. 3.2), we choose the weights for sub-region $i$ to be $\varphi_n^{(i)} = \frac{\exp((a\mathbb{I}_n^{(i)}+1)/\mathcal{T})}{\sum_{n=1}^{N} \exp((a\mathbb{I}_n^{(i)}+1)/\mathcal{T})}$, where $\mathbb{I}_n^{(i)}$ is an indicator variable that equals 1 if agent $n$ is assigned to explore $\mathcal{X}_i$ and equals 0 otherwise. We set $a = 15$ in all our experiments, and gradually increase the temperature $\mathcal{T}$ from 1 to $+\infty$. Specifically, for the synthetic experiments, we choose the temperature $\mathcal{T}$ as $\mathcal{T}_t = a/(a_t-1), \forall t \geq 1$; we set $a_t = a+1 = 16$ for the first 5 iterations ($t \leq 5$), decay the value of $a_t$ linearly to 1 in the next 5 iterations (i.e., $a_t = 16, 12.25, 8.5, 4.75, 1$ for $t = 6, \ldots, 10$), and fix $a_t = 1, \forall t > 10$. Note that when $a_t = 1$, $\mathcal{T}_t = +\infty$ (i.e., after 10 iterations) and the distribution becomes uniform among all agents. Similarly, for all real-world experiments, we use the same softmax weighting scheme except that we fix $a_t = a+1 = 16$ for the first 10 iterations, decay the value of $a_t$ linearly to 1 in the next 30 iterations, and fix $a_t = 1$ afterwards. That is, the distribution becomes uniform among all agents after 40 iterations. All our experiments are performed on a computing cluster where each device has one NVIDIA Tesla T4 GPU and 48 cores of Xeon Silver 4116 (2.1Ghz) processors.

## B.1  Synthetic Experiments

### B.1.1  Detailed Experimental Setting

Our synthetic experiments involve $N = 200$ agents. We define the domain of the synthetic functions to be 1-dimensional and discrete, i.e., an equally spaced grid on the 1-dimensional interval $[0, 1]$ with a domain size of $|\mathcal{X}| = 1000$. To generate the objective functions for the $N = 200$ different agents, we firstly sample a function $f$ from a GP with the SE kernel and a length scale of 0.03, and normalize the function values into the range $[0, 1]$. Next, for every agent $\mathcal{A}_n, \forall n \in [N]$, we go through all $|\mathcal{X}| = 1000$ inputs in the entire domain, and for each input $\mathbf{x}$, we derive the function value for agent $\mathcal{A}_n$ as $f^n(\mathbf{x}) = f(\mathbf{x}) + d$, in which $d = 0.02$ or $= -0.02$ with equal probability (i.e., a probability of 0.5 each). In this way, the objective functions of all agents are related to each other. When observing a function value, we add a Gaussian noise $\zeta \sim \mathcal{N}(0, \sigma^2)$ with a variance of $\sigma^2 = 0.01$ (Sec. 2).

To construct the $P$ sub-regions to be used for distributed exploration (DE), we simply need to divide the interval $[0, 1]$ into $P$ disjoint hyper-rectangles with equal volumes. For example, when $P = 2$, the two sub-regions contain the inputs in the sub-regions $[0, 0.5)$ and $[0.5, 1]$ respectively; when $P = 3$, the three sub-regions include the inputs in the sub-regions $[0, 1/3)$, $[1/3, 2/3)$ and $[2/3, 1]$ respectively.

### B.1.2  More Experimental Results

**Comparison between DP-FTS-DE and DP-FTS.** We have shown in the main text (Fig. 2a) that FTS-DE significantly outperforms FTS without DE. Here, we demonstrate in Fig. 4 that after DP is integrated, DP-FTS-DE still yields a significantly better utility than DP-FTS for the same level of privacy guarantee (loss). These results justify the practical benefit of the technique of DE (Sec. 3.2). Note that for a fair comparison, we have used a smaller value of $S$ for DP-FTS without DE such that a similar percentage of vectors are clipped for both DP-FTS-DE and DP-FTS.

**Investigation of DE.** We also investigate the importance of both of the major components of the DE technique (Sec. 3.2): (a) assigning every agent to explore only a local sub-region instead of the entire domain, and (b) giving more weights to those agents exploring the particular sub-region. In Fig. 5, the orange curve is obtained by removing component (b) (i.e., in every iteration and for each sub-region, we give equal weights to all agents), the purple curve is derived by removing component (a) (i.e., letting every agent explore the entire domain at initialization instead of a smaller local sub-region). As the figure shows, the performances of both the orange and purple curves are significantly worse than our FTS-DE algorithm (red curve), which verifies that both of these components are critical for the practical performance of FTS-DE.

**Trade-off Induced by $P$.** As we have discussed at the end of Sec. 4, the value of $P$ (i.e., the number of sub-regions) induces a trade-off about the practical performance of our DP-FTS-DE algorithm. Here we empirically verify this trade-off in Fig. 6. As shown in the figure, for the same values of $q$, $z$ and $S$, a smaller value of $P$ (i.e., larger local sub-regions) may deteriorate the performance (orange curve) since larger sub-regions are harder for the GP surrogate to model, however, a larger value of $P$

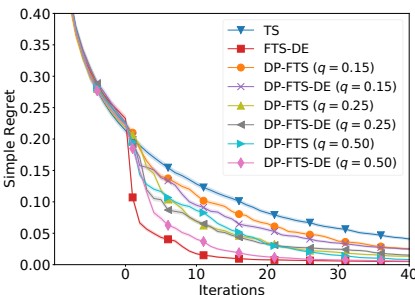

Figure 4: Comparisons of the performances of DP-FTS (without DE) and DP-FTS-DE. For a fair comparison, we have used $S = 8$ and $S = 11$ for DP-FTS and DP-FTS-DE respective, such that a similarly small percentage vectors are clipped for both algorithms ($0.31\%$ for DP-FTS and $0.80\%$ for DP-FTS-DE). We have used $z = 1.0$ for both algorithms.

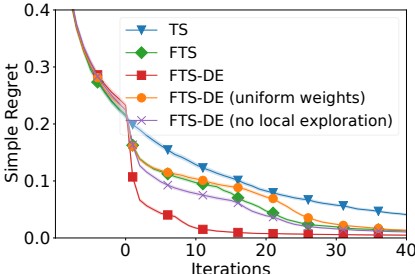

Figure 5: Investigating the importance of both major components of the technique of distributed exploration (DE). The orange curve is obtained by giving equal weights to all agent for every sub-region, and the purple curve is derived by letting every agent explore the entire domain at initialization instead of a local sub-region.

may also result in a worse performance (yellow curve) since it causes the vectors from more agents to be clipped (Sec. 4). These observations verify our discussions in the last paragraph of Sec. 4.

**Robustness Against Heterogeneous Agents.** We use another experiment to explore the robustness of our algorithm against agent heterogeneity, i.e., how our algorithm performs when the objective functions of different agents are significantly different. To begin with, we sample a function $f_{\text{base}}$ from a GP (the detailed setups such as the domain and the SE kernel are the same as those used in App. B.1.1). Next, for every agent $i = 1, \ldots, 50$, we independently sample another function $f^i$ and then use $f^i \leftarrow \alpha f^i + (1 - \alpha) f_{\text{base}}$ as the objective function for agent $i$ in which $\alpha \in [0, 1]$. As a result, the parameter $\alpha$ controls the difference between the objective functions $f^i$'s of different

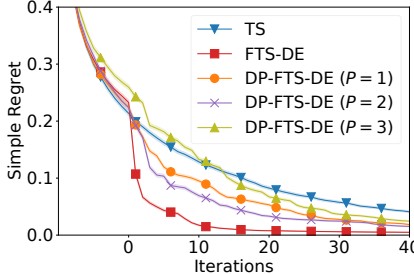

Figure 6: Trade-off induced by $P$ regarding the practical performance of our DP-FTS-DE algorithm. Note that a larger $P$ reduces the size of every local sub-region and hence leads to a better modeling by the GP surrogates, yet also negatively impacts the performance by causing more vectors to be clipped. Here we have used $q = 0.25, z = 1.0, S = 11.0$ for all values of $P$.

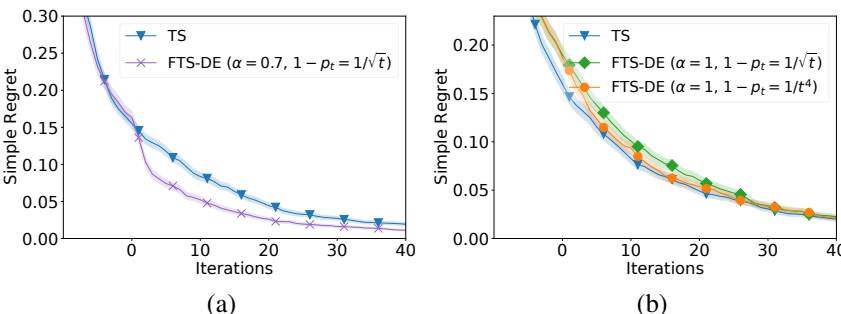

Figure 7: Results when the agents are heterogeneous, i.e., when the objective functions of different agents are significantly different. (a) and (b) correspond to $\alpha = 0.7$ and $\alpha = 1.0$, respectively.

agents such that a larger $\alpha$ means that the $f^i$'s are more different. Fig. 7a shows the performances of standard TS and our FTS-DE when $\alpha = 0.7$, in which FTS-DE is still able to outperform TS although the performance improvement is significantly smaller than that observed in Fig. 2. Fig. 7b plots the results when the objective functions $f^i$'s are extremely heterogeneous, i.e., when $\alpha = 1.0$ which implies that the $f^i$'s are *independent*. The figure shows that in this adverse scenario, when $1 - p_t = 1/\sqrt{t}$, FTS-DE (green curve) performs worse than standard TS (blue curve) due to the extremely high degree of heterogeneity among the agents. However, as shown by the orange curve, we can improve the performance of FTS-DE to make it comparable with standard TS by letting $1 - p_t$ decrease faster such that the impact of the other agents are diminished faster.

## B.2 Real-world Experiments

### B.2.1 More Experimental Details

In all real-world experiments, when generating the random features for the RFFs approximation, we use the SE kernel with a length scale of $0.01$ and a variance of $\sigma^2 = 10^{-6}$ for the observation noise. Refer to [12] and [53] for more details on how the random features are generated and how they are shared among all agents.

As we have mentioned in the main text, we use $P = 4$ sub-regions in all three real-world experiments, and divide the entire domain into $P = 4$ hyper-rectangles (i.e., sub-regions) with equal volumes. Following the common practice in BO, we assume that the domain $\mathcal{X} \in \mathbb{R}^D$ is a $D$-dimensional hyper-rectangle, and w.l.o.g., assume that every dimension of the domain is normalized into the range $[0, 1]$. That is, the domain can be represented as $[0, 1]^D = \{[0, 1], [0, 1], \ldots, [0, 1]\}$. Note that every domain which is a hyper-rectangle can be normalized into this form. As a result, when the input dimension is $D = 2$ (i.e., the landmine detection experiment), we construct the $P = 4$ hyper-rectangles such that $\mathcal{X}_1 = \{[0, 0.5), [0, 0.5)\}$, $\mathcal{X}_2 = \{[0, 0.5), [0.5, 1.0]\}$, $\mathcal{X}_3 = \{[0.5, 1.0], [0, 0.5)\}$ and $\mathcal{X}_4 = \{[0.5, 1.0], [0.5, 1.0]\}$. Similarly, when the input dimension $D = 3$ (i.e., the human activity recognition and EMNIST experiments), we construct the $P = 4$ hyper-rectangles such that $\mathcal{X}_1 = \{[0, 0.5), [0, 0.5), [0, 1]\}$, $\mathcal{X}_2 = \{[0, 0.5), [0.5, 1.0], [0, 1]\}$, $\mathcal{X}_3 = \{[0.5, 1.0], [0, 0.5), [0, 1]\}$ and $\mathcal{X}_4 = \{[0.5, 1.0], [0.5, 1.0], [0, 1]\}$.

The *landmine detection* dataset[15] used in this experiment has also been used by the works of [12, 60] which focus on FBO and FL respectively. This dataset consists of the landmine detection data from $N = 29$ landmine fields (agents), and the task of every agent is to use a support vector machine (SVM) to detect (classify) whether a location in its landmine field contains landmines or not (i.e., binary classification). We tune two hyperparameters of SVM, i.e., the RBF kernel parameter ($[0.01, 10.0]$) and the penalty parameter ($[10^{-4}, 10.0]$). For every landmine field, we use half of its dataset as the training set and the remaining half as the validation set. In this experiment, we report the area under the receiver operating curve (AUC) as the performance metric, instead of validation error, because this dataset is significantly imbalanced, i.e., the vast majority of the locations do not contain landmines. No data is excluded. Refer to [66] for more details on this dataset. The dataset is publicly available, and contains no personally identifiable information or offensive content.

---

[15]http://www.ee.duke.edu/~lcarin/LandmineData.zip.

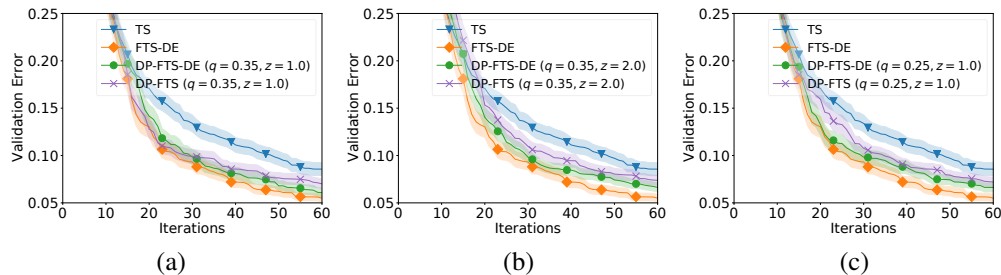

Figure 8: Comparison between DP-FTS and DP-FTS-DE for the same level of privacy guarantee (the human activity recognition experiment). We have used $S = 22$ and $S = 11$ for DP-FTS-DE and DP-FTS (without DE) respectively, such that a similar percentage of vectors are clipped in both cases: $1.02\%$ for DP-FTS-DE and $1.09\%$ for DP-FTS.

The *human activity recognition* dataset[16] was originally introduced by the work of [2] and has also been adopted by the works of [12, 60]. The licensing requirement for this dataset requires that the use of this dataset in publications must be acknowledged by referencing the paper of [2]. This dataset consists of the data collected using mobile phone sensors when $N = 30$ subjects (agents) are performing six different activities. The task of every agent (subject) is to use the dataset generated by the subject to perform activity recognition, i.e., to predict which one of the six activities the subject is performing using logistic regression (LR). We tune three hyperparameters of LR: the batch size ($[128, 512]$), L2 regularization parameter ($[10^{-6}, 10]$) and learning rate ($[10^{-6}, 1]$). For every subject, we again use half of its data as the training set and the other half as the validation set, and the validation error is reported as the performance metric. The inputs for every agent are standardized by removing the mean and dividing by the standard deviation of its training set, which is a common pre-processing step for LR. No data is excluded. Refer to [2] for more details on this dataset. The dataset is publicly available as described above, and does not contain personally identifiable information or offensive content.

*EMNIST*[17] is a dataset of images of handwritten characters from different persons, and is a widely used benchmark in FL [29]. The EMNIST dataset is under the CC0 License. Here we use the images from the first $N = 50$ subjects (agents) which can be accessed from the TensorFlow Federated library[18]. Every subject (agent) uses a convolutional neural network (CNN) to learn to classify an image into one of the ten classes corresponding to the digits $0 - 9$. Here the task for every agent is to tune three CNN hyperparameters: learning rate, learning rate decay and L2 regularization parameter, all in the range of $[10^{-7}, 0.02]$. We follow the standard training/validation split offered by the TensorFlow Federated library for every agent, and again use the validation error as the performance metric. All images are pre-processed by normalizing all pixel values into the range of $[0, 1]$, and no data is excluded. Refer to [8] for more details on this dataset. The dataset is publicly available, and contains no personally identifiable information or offensive content.

### B.2.2 Comparison between DP-FTS-DE and DP-FTS

We have shown in the main text (Figs. 3a,b,c) that FTS-DE significantly outperforms FTS without DE. Here we further verify in Fig. 8 the importance of the technique of DE after DP is integrated, using the human activity recognition experiment. Specifically, the figures show that after the incorporation of DP, DP-FTS-DE (green curves in all three figures) still achieves a better utility than DP-FTS (purple curves) for the same level of privacy guarantee (loss). Note that same as Fig. 4, to facilitate a fair comparison, we have used a smaller value of $S$ for DP-FTS without DE such that a similar percentage of vectors are clipped for both DP-FTS-DE and DP-FTS.

---

[16]https://archive.ics.uci.edu/ml/datasets/Human+Activity+Recognition+Using+Smartphones.

[17]https://www.nist.gov/itl/products-and-services/emnist-dataset.

[18]https://www.tensorflow.org/federated.

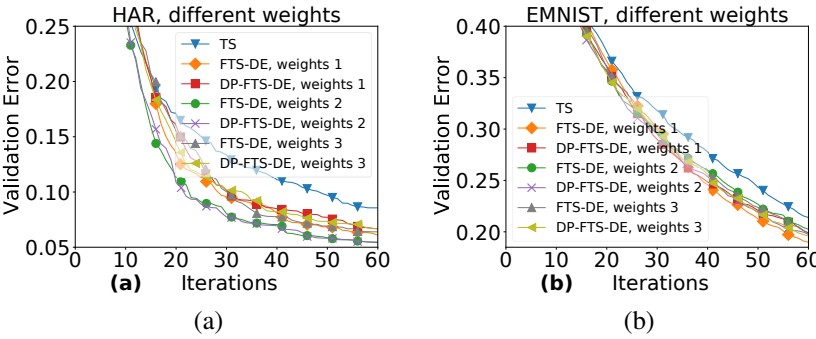

(a)                                     (b)

Figure 9: Robustness of our FTS-DE and DP-FTS-DE algorithms against the choice of weights.

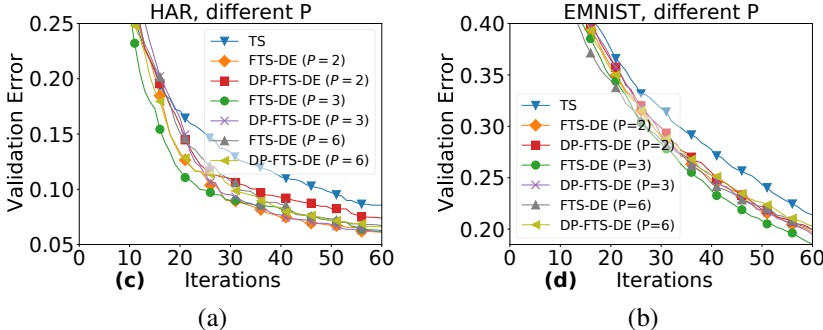

(a)                                     (b)

Figure 10: Robustness of our FTS-DE and DP-FTS-DE algorithms against the number $P$ of sub-regions.

### B.2.3   Robustness against the Choice of Weights and Number of Sub-regions

In this section, we evaluate the robustness of our experimental results against the choice of the weights assigned to different agents and the number $P$ of sub-regions, using the human activity recognition and EMNIST experiments.

Here we test three other methods for designing the weights: (1) We use the same softmax weighting scheme with a different parameter $a = 9$ instead of $a = 15$ ("weights 1" in Fig. 9). (2) For a sub-region $\mathcal{X}_i$, we assign a weight $\propto a$ to those agents exploring $\mathcal{X}_i$ and $\propto b$ to the other agents, and similarly gradually decay the value of $a = 1,000$ to $b = 1$ ("weights 2" in Fig. 9). (3) For a sub-region $\mathcal{X}_i$, we assign a weight $\propto a^2$ to those agents exploring $\mathcal{X}_i$ and $\propto b$ to the other agents, and similarly gradually decay the value of $a = 40$ to $b = 1$ ("weights 3" in Fig. 9). Moreover, in addition to the results using $P = 4$ sub-regions reported in the main text, here we also evaluate the performance of FTS-DE and DP-FTS-DE with $P = 2, 3, 6$ sub-regions (Fig. 10). All DP-FTS-DE methods in Fig. 9 and Fig. 10 correspond to $q = 0.35, z = 1.0$. The results demonstrate the robustness of our experimental results against the choice of the weights and the number $P$ of sub-regions.

### B.2.4   Rényi DP

Fig. 11 shows the privacy-utility trade-off in the landmine detection experiment using Rényi DP [63]. The results demonstrate that Rényi DP, despite requiring modifications to our theoretical analysis (i.e., proof of Theorem 1), leads to slightly better privacy losses (compared with Fig. 3a) with comparable utilities.

### B.2.5   Adaptive Weights vs. Non-adaptive Weights

As we have discussed in the last paragraph of Sec. 3.2, we have designed the set of weights for every sub-region to be adaptive such that they gradually become uniform among all agents as $t$ becomes large. Here we explore the performance of our algorithm if the weights are *non-adaptive*, i.e., for every sub-region $\mathcal{X}_i$, we fix the set of weights $\{\varphi_n^{(i)}, \forall n \in [N]\}$ for all $t \in [T]$. In particular, we

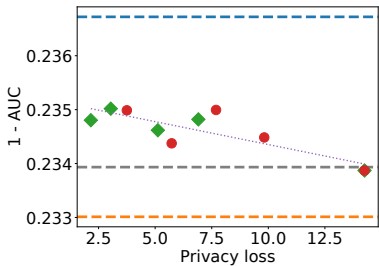

Figure 11: Results for the landmine detection experiment using Rényi DP [63].

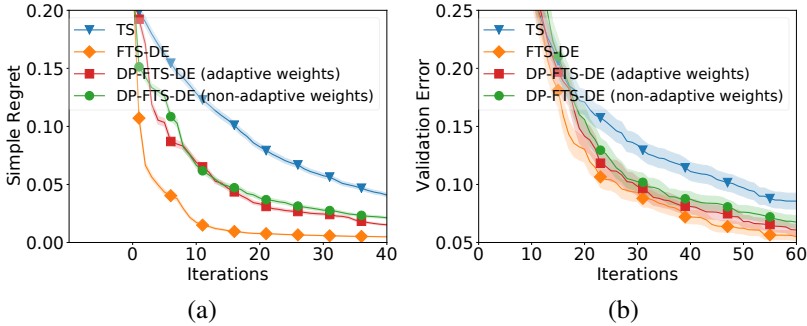

Figure 12: Comparison between DP-FTS-DE with adaptive weights and non-adaptive weights, using (a) the synthetic experiment and (b) human activity recognition experiment.

adopt the same softmax weighting scheme as described in the first paragraph of App. B, but fix the temperature $\mathcal{T}_t = 1, \forall t \geq 1$ such that the same set of weights is used for all $t \geq 1$. That is, for every sub-region $\mathcal{X}_i$, we assign more weights to those agents exploring $\mathcal{X}_i$ throughout all iterations $t \geq 1$.

Fig. 12 shows the comparisons between adaptive and non-adaptive weights using (a) the synthetic experiment and (b) human activity recognition experiment. Both figures show that although in the initial stage, DP-FTS-DE with non-adaptive weights performs similarly to DP-FTS-DE with adaptive weights, however, as $t$ becomes large, adaptive weights (red curves) lead to better performances than non-adaptive weights (green curves). This can be attributed to the fact that as $t$ becomes large, every agent is likely to have explored (and become informative about) more sub-regions in addition to the sub-region that it is assigned to explore at initialization. Therefore, if the weights are non-adaptive, i.e., for a sub-region $\mathcal{X}_i$, if after $t$ has become large, most weights are still given to those agents that are assigned to explore $\mathcal{X}_i$ at initialization, then *the information from the other agents* who are likely to have become informative about $\mathcal{X}_i$ (i.e., have collected some observations in $\mathcal{X}_i$) *is not utilized*. This under-utilization of information might explain the performance deficit caused by the use of non-adaptive weights. However, note that despite being outperformed by DP-FTS-DE with adaptive weights, DP-FTS-DE with non-adaptive weights (green curve) is still able to consistently outperform standard TS (blue curves).

### B.2.6 Computational Cost

When maximizing the acquisition function to select the next query (lines 6 and 8 of Algo. 2), firstly, we uniformly randomly sample a large number (i.e., 1000) of points from the entire domain; next, we use the L-BFGS-B method with 20 random restarts to refine the optimization.

For the central server, the integration of DP and DE (in expectation) incurs an additional computational cost of $\mathcal{O}(PNq)$. However, these additional computations are negligible since they only involve simple vector additions/multiplications (lines 5-11 of Algo. 1). For agents, the incorporation of DP brings no additional computational cost to them. Meanwhile, the addition of DE, which affects line 8 of Algo. 2, only incurs minimal additional computations. For example, in the landmine detection experiment, line 7 takes on average 14.47s and 17.96s to compute for $P = 1$ and $P = 4$, respectively.