# OpenReview forum: "Differentially Private Federated Bayesian Optimization with Distributed Exploration"
_NeurIPS.cc/2021/Conference — NeurIPS 2021 Poster_

### Official Review · Reviewer_YPjs · 2021-07-08

**Rating:** 7
**Confidence:** 3

**Summary:**

This paper proposes an algorithm for performing Federated Bayesian Optimisation using differential privacy—differentially private federated Thompson sampling. Differential Privacy is maintained by aggregating local updates using a subsampled Gaussian mechanism. Additionally, distributed exploration is applied to warm-start the algorithm, and allows for more efficient exploration, especially in the earlier iterations.

One sentence summary: the authors propose DP-Federated BO using Thompson sampling with a distributed initial random search.

**Ethical Concerns:**

I do not have ethical concerns with this paper.

**Limitations And Societal Impact:**

Unfortunately, there seems to be only a token engagement with the broader societal implications of the work and the limitations of the work.

**Main Review:**

Please note that I have organised my points in this review by category, and also whether they are a strength of a weakness of the work. Points marked * I deem to be especially important, and should be addressed in the author response if the authors would like me to reconsider my score.

EDIT: I have increased my score to 7 based on the authors response.

---
- Originality (O)
    - (+O1) The authors primarily combine known techniques in the DP and BO literature to propose this algorithm. Of course, some modifications are necessary. The combination of these ideas is well executed, and a strength of the paper.

---
- Quality (Q)
    - (+Q1) I was impressed by the theoretical analysis of the regret of the DP-FTS-DE algorithm. The conclusions from the theory match intuition, and it is reassuring to see this reflected in the theory. Note that however I was unable to verify the "correctness" of the theory, because it is far from my area of expertise.
    - (Q2) The analysis assumes that all objective functions have a bounded normal induced by the RKHS associated with kernel k. Please comment on whether this is a key limitation of the work.
   - (Q3) The theoretical analysis hinges on d_n, which quantifies the similarity in the objective functions between clients. Is this known in practice? We are performing Bayesian Optimisation because function evaluation is difficult.
    - (-Q4)** There is insufficient discussion of the threat model and the specific setting that this algorithm can be applied in. Specific questions that I have regarding the settings are: (i) which data is considered private, and which data is considered public? Or is all data considered public (ii)** i assume that leaking f_i(x) would leak data relating to client i only, because otherwise the analysis of the algorithm would be incorrect. For example, if f_i(x) is the validation accuracy when training on the data from all clients (and each client has the same f_i(x)), when releasing f_i(x) actually leaks information about **all** users, rather than just one, and the true privacy protection is significantly diminished. Perhaps I missed discussion of this point, but I believe that the assumptions made are critical. (iii) Is it assumed that every client is able to update and send parameter updates simultaneously? Often in FL, clients have different network capabilities. (iv) Who is trusted, and who is not trusted? Line 148 "FTS requires an agent to receive all vectors from other agents so the transformations requires by the general DP framework cannot be easily incorporated". This is assuming that other clients are not trustworthy; this should be made clear in the problem setting. (v) What are adjacent datasets considered to be, in terms of defining DP? Considering the algorithm description, adjacent datasets seem to be datasets where all of the data on a specific client varies. This might be excessive, if for example, a "client" is actually a hospital, and the data at that client is actually from several different individuals.
    - (-Q5)* Information from different clients is combined using a weighted average, where the weight is determined by whether a particular client was "assigned" an area or search initially. The weighted average does not seem principled to me. How would it compare to, for example, using a Gaussian Process on all of the clients data? How are we accounting for how each client has their own f_i, and thus the value of f_i at client m may not be representative of the the value of f_i at client n. Ideally we would allow for uncertainty in how the values of f may different across clients.
    - (-Q6) Proposition 1 shows that the algorithm is DP. However, I think it would be better to make clear exactly how the privacy loss was computed (i.e., the moments accountant), rather that state a theorem that is effectively identical to that of Abadi et al.
    - (-Q7)* The authors have shown that the algorithm is asymptomatically no regret. As I understand, this means that, over time, the algorithm will select closer to optimal actions. However, the constants matter. What happens if the f_i are chosen very different for each client? While this algorithm is asymptotically no-regret, how does it compare to simply performing local Bayesian Optimisation for each client? This could be evaluated experimentally, by adding a synthetic example where the f_i are "significantly" different across clients, and comparing local BO to federated BO.
    - (Q8) The first term in Theorem 1 is sub-linear for the squared exponential kernel (see Line 253). Are there kernels for which this term is not sublinear?
    - (+Q9) As I understand, the value of distributed exploration is particularly high in the few iteration regime, such as when there is a strict privacy budget. Using distributed exploration in the differential privacy context is a good combination.
   - (-Q10)* In Figure 3, only 60 iterations are run, even though the algorithms have clearly not converged. Why did you stop at 60 iterations? The experiments should be run with more experiments.
   - (-Q11)* I was disappointed to see that authors did not engage with the limitations of the work. Perhaps thinking carefully about the specific context this algorithm can be applied in (See Q4) may reveal some limitations e.g., certain settings that the algorithm cannot be applied.
---
Overall, I believe improving the clarity and precision of the text in the paper would greatly benefit the submission.

- Clarity (C)
    - (-C1) Additional clarity could be provided about the Federated Bayesian Optimisation setting. For example, Introduction line 20-22 "BO has recently been extended to .... without requiring them to share their raw data" could be interpreted as parallel Bayesian optimisation, where there is only one single objective function.
    - (-C2) The authors write (lines 43/44) that "DP-FedAvg guarantees that an adversary ... cannot infer whether data from a particular user has been used by the algorithm", is too strong. DP-FedAvg often uses epsilon, delta differential privacy, and therefore there is a delta chance (ie non-zero) of their being potentially catastrophic privacy breach.
    - (-C3)* The authors write that "every agent can explore the local sub-region more effectively because its GP surrogate can model the objective function more accurately in a smaller local-subregion" (lines 60-62). My understanding of reference [16] is that better local perform is due to being able to fine-tune parameters locally i.e., for each subregion. Is that what is happening here? Or does this approach ensure better that the full search space is explored e.g., by having users not search in the same area. Please clarify. Later, the authors write "smaller sub-regions are easier to model for the GP surrogate, and hence can be better explored" (line 312/313)—is this specific to GPs? Is this some property of inference?.
    - (-C4) Line 82 "An interesting trade-off between privacy and utility in real-world applications"—what does this mean?
    - (-C5) The algorithm suggested in the paper builds heavily on the Federated Thompson Sampling algorithm, but this algorithm is not explained in detail in the background section. Ideally, much more space would be devoted to this prior work. For example, intuition on why randomly sampling the optimal point from another agent is a good idea.
    - (-C6) As I understand, the privacy loss is calculated using the moments accountant. I suggest adding a line similar to "update privacy loss using moment accountant" to the algorithm box. This point should be made clearer.
    - (-C7) I think lines 251 to 300 could be written in a more reader friendly way e.g., by reminding the reader of what certain symbols mean.
    - (-C8) Figure 2 is hard to read, and not colorblind friendly.

---
- Significance (S)
    - (S1) The specific combination of ideas in this paper, to my knowledge, are novel, and could be used in practice. On the other hand, it is unclear to me how others in the community will be able to build upon this work. Since this work is the first to consider federated, private BayesOpt, the results are state of the art.

---
- Miscellaneous (M)
    - (M1) Abstract Line 14 "competitive performance"—how is this measured? Relative to non-private performance?
    - (M2) Abstract Line 15 "practical trade-off between privacy and utility". Please clarify. Are you stating that the algorithm performs "well" and achieves a favourable trade-off, or that the algorithm allows for trading off privacy and utility e.g., through choice of hyperparameter.
    - (M3) Line 56 "elegant combination, please omit elegant.
    - (M4) What is the reason for the instability in results in Figure 3(c)—increasing the privacy budget/loss sometimes degrades performance?


**Time Spent Reviewing:**

5

---

> ### Author Response · Authors · 2021-08-10
> **Response to Reviewer YPjs**
>
> We would like to thank the reviewer for your thoughtful and detailed feedback. We will follow your suggestions carefully to improve our paper.
>
> ***
>
> **Quality (Q)**
> * (Q2) The assumption that an objective function has a bounded norm induced by the RKHS associated with a kernel k is a common smoothness assumption in the analysis of BO algorithms [6]. Our assumption that all objective functions have such a bounded norm (which is also adopted by [10]) is also not restrictive because it essentially corresponds to a smoothness assumption on the most unsmooth objective function among all clients. In particular, for the squared exponential (SE) kernel which we focus on in this work (line 91), the kernel k is simply the SE kernel with the smallest length scale among all clients. Therefore, this assumption is not a limitation of this work.
> * (Q3) The value of $d_n$ is unknown both in theory and in practice.
> * (Q4) (i) All data held by every client is considered private.
>
>     (ii) $f^i(x)$ is the validation accuracy **when training only on the data of client $i$**. So, the reviewer is correct that **leaking $f^i(x)$ would leak information relating to client $i$ only** and does not leak information about other users.
>
>     (iii) In our algorithmic design and theoretical analysis, we have assumed that every client is able to update and send parameter updates simultaneously in every iteration. In practice, a simple heuristic can be used to handle the difference in network capabilities. That is, the central server can modify the weights (i.e., $\varphi^{(i)}_n$’s) assigned to different clients such that it assigns a weight of 0 to those clients whose parameters are not received in the current round.
>
>     (iv) As specified in Footnote 2 at the bottom of page 2, we assume that the central server is trustworthy following the work of [28], and the reviewer is correct that the clients are considered not trustworthy in our setting. We will revise and make these more explicit.
>
>     (v) The reviewer is correct that we define two datasets as adjacent if they differ by all the data from a single client/user (lines 123-124). This definition is the same as the one adopted by DP-FedAvg [28] (Definition 2 in [28]) and it is reasonable because, same as DP-FedAvg [28], we are interested in preserving the **user-level** privacy. That is, we aim to prevent an adversary from inferring whether (all the data from) a client has participated in the algorithm.
>
> * (Q5) In our theoretical analysis, the differences between the objective functions $f^i$s of different clients are characterized by the parameters $d_n$’s (lines 111-113). As a result, the weighted average is a reasonable choice because it leads to a theoretical guarantee (Theorem 1) that explicitly accounts for the differences between the objective functions of different clients, i.e., our regret bound in Theorem 1 explicitly depends on $d_n$’s. Using a Gaussian Process on all of the clients data is infeasible because (a) the objective functions of different clients can be (potentially highly) different and (b) sharing the clients raw data is not allowed in the federated setting.
>
> * (Q6) The calculation of the privacy loss is briefly mentioned in Footnote 4 at the bottom of page 4. We will follow your suggestion and add more details on it.
>
> * (Q7) When the $f^i$s of the clients are significantly different from each other, the performance of our algorithm inevitably becomes worse. However, as reflected by our theoretical result which guarantees that our algorithm is asymptotically no-regret regardless of the differences between the $f^i$s, our algorithm is still able to perform comparably with standard TS even in this adverse scenario. We have corroborated this by adding a synthetic experiment following your suggestion.
>
>     The basic setting of this added experiment is similar to our synthetic experiments in Section 5.1. We firstly sample a function $f_{\text{base}}$ from a GP. Next, for each agent $i=1,\ldots,50$, we independently sample another function $f^i$ and use $f^i \leftarrow \alpha f^i + (1-\alpha) f_{\text{base}}$ as the objective function of agent $i$. As a result, the parameter $\alpha\in[0,1]$ controls the difference between the objective functions $f^i$s of different agents, such that a larger $\alpha$ means the $f^i$s are more different. The table below shows the simple regret after different numbers of iterations for $\alpha = 0.7$ (with $1-p_t=1/\sqrt{t}$), in which FTS-DE still outperforms standard local TS.
>
>    | Method | $10$ iter | $20$ iter | $30$ iter | $40$ iter |
>    | :--- | :--- | :--- | :--- | :--- |
>    | TS      | $0.084$ | $0.044$ | $0.027$ | $0.019$ |
>    | FTS-DE | $0.052$ |  $0.026$ | $0.016$ | $0.011$ |
>
>     The table below presents the results in the extreme case where the $f^i$s are significantly different i.e., when $\alpha = 1$ which implies that the $f^i$s are independent. The table shows that in the default setting with $1-p_t=1/\sqrt{t}$, FTS-DE now performs worse than local TS due to the extremely high degree of heterogeneity among agents. However, by letting $1-p_t$ decrease faster (i.e., by diminishing the impact of the other agents faster), we can improve the performance of FTS-DE to make it comparable with standard local TS.
>
>    | Method | $10$ iter | $20$ iter | $30$ iter | $40$ iter |
>    | :--- | :--- | :--- | :--- | :--- |
>    | TS      | $0.083$ | $0.049$ | $0.031$ | $0.020$ |
>    | FTS-DE ($1-p_t=1/\sqrt{t}$) | $0.101$ |  $0.060$ | $0.034$ | $0.022$ |
>    | FTS-DE ($1-p_t=1/t^4$) | $0.093$ |  $0.054$ | $0.033$ | $0.021$ |
>
>     We will add this additional experiment, as well as the discussions here, to the paper.
>
> * (Q8) The first term in Theorem 1 may not be sub-linear for the Matern kernel [6]. Note that the dependence of the first term on $T$ is the same as the standard BO algorithm using TS [6].
>
> * (Q10) Following your suggestion, we have extended Fig. 3 (e) (the human activity recognition experiment) to 200 iterations, and the results are consistent as shown in the table below. The results show that after 200 iterations, all methods have converged to very similar values. We will also extend the other experiments to 200 iterations and add the results to the paper after revision.
>
>    | Method | $100$ iterations | $150$ iterations | $200$ iterations |
>    | :--- | :--- | :--- | :--- |
>    | TS      | $0.062 \pm 0.005$ | $0.042 \pm 0.003$ | $0.032 \pm 0.002$ |
>    | FTS-DE | $0.040 \pm 0.003$ |  $0.033 \pm 0.002$ | $0.027 \pm 0.001$ |
>    | DP-FTS-DE ($q=0.35,z=1.0$) | $0.046 \pm 0.003$ | $0.036 \pm 0.002$ | $0.029 \pm 0.002$ |
>    | DP-FTS-DE ($q=0.35,z=2.0$) | $0.050 \pm 0.004$ | $0.037 \pm 0.002$ | $0.028 \pm 0.002$ |
>    | DP-FTS-DE ($q=0.25,z=1.0$) | $0.050 \pm 0.004$ | $0.036 \pm 0.002$ | $0.029 \pm 0.002$ |
>
> * (Q11) Regarding the limitations of our work, firstly, we will add more discussions on the limitation we have discussed in Section 7 regarding more advanced privacy-preserving techniques (lines 365-367), which we originally did not expand on due to lack of space. The more advanced privacy-preserving technique of Gaussian DP delivers smaller privacy losses than the moments accountant method (which we have adopted in this paper), and has been widely applied in practice (for example, Gaussian DP has been included in Tensorflow Privacy which is a widely used library for privacy-preserving ML). This makes the standard moments accountant method slightly outdated. However, incorporating Gaussian DP into our algorithm requires modifications to our theoretical analysis (i.e., the proof of Theorem 1) which may be challenging.
>
>     Moreover, thanks to your insightful comment in Q4 (iii), we have recognized another limitation of our current algorithm: we do not have a principled way to account for the different network capabilities of different clients, which is a common problem in the federated setting. Accounting for this issue would require modifications to our algorithmic design (and hence theoretical analysis) and is an interesting topic for future works. We will also add this limitation to the paper after revision.
>
> ***
>
> **Clarify (C)**
> * (C3) In reference [16], the use of local GPs is inspired by the classic trust-region method, which has shown that **small local regions can be modelled more accurately by the surrogate model** (see Section 2, Subsection “Local modeling” in [16]). A good local performance in [16] is achieved by adapting the size of the trust region (TR) such that the TR is (a) **small enough such that the GP surrogate can model the objective function accurately** and (b) large enough to contain good inputs (see Section 2, Subsection “Trust regions” in [16]). Based on the same principle of local modeling, our DE technique divides the entire search space into smaller local sub-regions and assigns every agent to explore only one local sub-region during initialization. As a result, **every local sub-region can be modelled more accurately by the GP surrogates** of the corresponding agents. The sentence in lines 312/313 in fact refers to the advantage of local modeling in general. Therefore, it is not specific to GPs, but is applicable to any surrogate model.
> * (C1, C2, C4-C8) We will also follow these suggestions to improve the clarity of our paper.
>
> ***
>
> **Miscellaneous (M)**
> * (M1) By “competitive performance”, we mean that our algorithm performs better than standard local BO and competitively with non-private FTS-DE. We will clarify this.
> * (M2) What we mean here is that our algorithm allows trading off privacy and utility. We will revise this part to clarify this.
> * (M3) We will remove “elegant” as you have suggested and also revise the paper to fix similar issues.
> * (M4) The instability in results in Figure 3(c) is due to the randomness in the experiments. In particular, it is because there are many sources of randomness in our experiments, such as the subsampling step, the addition of Gaussian noise, etc.

---

### Official Review · Reviewer_bP1K · 2021-07-16

**Rating:** 7
**Confidence:** 3

**Summary:**

The paper presents a differentially private FTS with distributed exploration. The contributions include (1) the development of the algorithm; (2) theoretical analysis of its privacy-utility tradeoff; and (3) experimental evaluation.

**Limitations And Societal Impact:**

Yes, the authors addressed the limitations

**Main Review:**

Pros:
- the paper is nicely structured and well-written
- theoretical analysis for the proposed algorithm on utility is solid
- empirical study is presented to validate the effectiveness of the proposed algorithm

Con:
- the proposed algorithm introduces some additional hyperparameters like #/ exploration regions, exploration temperature on top of DP-FTS. They need to be tuned to get good performance. but the tuning of these parameters will introduce additional privacy loss. It'll be good if the authors can provide a discussion of tuning the parameters and their privacy implications.


**Time Spent Reviewing:**

2

---

> ### Author Response · Authors · 2021-08-10
> **Response to Reviewer bP1K**
>
> We highly appreciate the reviewer for acknowledging our contributions.
>
> ***
>
> Con:
> * Regarding the additional hyperparameters such as the number of sub-regions and the temperature parameter, our algorithm can use a single set of hyperparameters that yields consistent performances across different applications, which removes the need to tune these hyperparameters for every new use case. This is supported by our experimental results because we have applied the same set of hyperparameters (including the number of sub-regions, the temperature parameter, etc.) in all three real-world experiments (Section 5.2) and they have consistently produced competitive performances (Figure 3).
>
>     In the practical deployment of our algorithm, we can firstly use simulation experiments or an initial set of real-world experiments to tune the values of these hyperparameters, during which the additional privacy loss can be easily accounted for using commonly used techniques (e.g., the method adopted by DP-SGD, described in Section 3.3 of [1]). After that, we can apply these tuned hyperparameters to new similar use cases, which will incur no additional privacy loss. When the use case is significantly different from those we have encountered before, we can always resort to commonly used techniques (such as the method adopted by DP-SGD) to account for the additional privacy losses.

---

### Official Review · Reviewer_7UgD · 2021-07-16

**Rating:** 7
**Confidence:** 3

**Summary:**

This paper prosed the first algorithm with a rigorous guarantee on the user-level privacy in the federated bayesian optimization setting. The proposed method integrated differential privacy federated Thompson sampling. Many theoretical insights of the privacy-utility trade-off are given, and the performance of proposed algorithm are validated on real-world experiments.


**Ethics Review Area:**

["I don’t know"]

**Limitations And Societal Impact:**

Yes

**Main Review:**

Strength:
1. The paper are well written and presented, and the problem is very important.
2. The technical parts seem solid. It is good to see that a differentially private federated learning algorithm can handle the heterogeneity of local devices.

Weakness:
No major weaknesses. This is a strong submission.

**Time Spent Reviewing:**

1 hour

---

> ### Author Response · Authors · 2021-08-10
> **Response to Reviewer 7UgD**
>
> We would like to thank the reviewer for appreciating the contributions of our paper.

---

### Official Review · Reviewer_wuG4 · 2021-07-19

**Rating:** 6
**Confidence:** 4

**Summary:**

This work suggests a differentially private federated Bayesian optimization framework, which is extended Bayesian optimization to federated learning. Following the previous work [1], this work optimizes an objective in respective agents and occasionally updates the information obtained from the agents to a central server. However, [1] did not guarantee a rigorous privacy, which is the most important part of federated learning. By leveraging a differential privacy concept and distributed exploration, the authors propose differentially private federated Thompson sampling with distributed exploration (DP-FTS-DE), which theoretically guarantees a sublinear regret. Finally, the authors demonstrate that the proposed method outperforms other baselines including Thompson sampling and FTS-DE.

**Ethical Concerns:**

I do not have any ethical concerns.

**Limitations And Societal Impact:**

I do not have any comments on limitations and societal impact. Please see "Main Review" for the detailed reviews.

**Main Review:**

This paper solves an interesting topic, which can be widely adopted in a practical scenario. In particular, as a real-world problem, a federated learning scheme is able to preserve a privacy information as well as solve a target task in each agent. In this paper, from the idea of online regression with random Fourier features, it communicates the coefficients between central server and agents by incorporating the concepts of distributed exploration and differential privacy. However, I have some concerns on assumptions and numerical experiments. Please read the comments described below and provide a response in the rebuttal.

Pros

+ Interesting topic.
+ Well-written and well-organized paper.

Cons

- Limited novelty.
- Unrealistic assumptions.
- Weak numerical experiments.

Detailed Comments

1. Could you provide an intuitive explanation on subsampled Gaussian mechanism at the beginning of Section 3.3? If this is added, a potential reader would understand well about the concept of subsampled Gaussian mechanism.
2. I do not want to point out the novelty of this work directly, but the components in the proposed method have been already suggested in the existing studies [1, 2]. If I misunderstood, please let me know in the rebuttal.
3. Importantly, I think the choice of $1 - p_t$ is unrealistic. If $1 - p_t \in \mathcal{O}(1/t^2)$, Line 7 of Algorithm 2 would not be operated with a very high probability after a few iterations. This is not the authors' intention for the proposed method. Moreover, it implies that the regret bound becomes almost same to the standard GP-UCB.
4. Related to the comment above, in the experiments, the authors choose a slowly decreasing $1 - p_t$ such as $\mathcal{O}(1/\sqrt{t})$. Such a choice makes a regret bound not hold. I would like to be informed about any thoughts on this.
5. For Figure 3, Bayesian optimization with 60 iterations is not enough, because they do not converge yet.

Minor issues

* Could you provide a reason why there are the experiment results before 0 iteration in Figure 2?
* For Algorithm 2, the part from Line 4 to Line 7 should be declared by if-else statement.
* In Line 84, what is $\mathcal{R}$? It seems like $\mathbb{R}$. There exist wrong notations like this. Please fix them.
* In Line 18, DNN should be DNNs.
* In Line 99, RFF should be RFFs.

[1] Z. Dai, K. H. Low, and P. Jaillet. Federated Bayesian optimization via Thompson sampling. In Advances in Neural Information Processing Systems (NeurIPS), volume 33, Virtual, 2020.

[2] Z. Ji, Z. C. Lipton, and C. Elkan. Differential privacy and machine learning: a survey and review. arXiv preprint arXiv:1412.7584, 2014.

**Time Spent Reviewing:**

5

---

> ### Author Response · Authors · 2021-08-10
> **Response to Reviewer wuG4**
>
> We would like to thank the reviewer for your valuable feedback, which we will follow carefully to improve our paper.
>
> ***
>
> Detailed Comments:
> 1. The subsampled Gaussian mechanism has been introduced in Section 2 Background (lines 130-315). We will follow your suggestion and add a brief explanation of it at the beginning of Section 3.3 to improve understanding.
> 2. Although the FTS algorithm and the general DP framework have been separately suggested in existing studies, one of our major novel contributions in this work in fact lies in the **theoretical analysis (Sec. 4) and empirical demonstration (Sec. 5) of the privacy-utility trade-off** of our proposed DP-FTS-DE algorithm, which are non-trivial and have never been done in existing studies. Furthermore, another major novel contribution of this work is the method of **distributed exploration (DE)** (Sec. 3.2), which arises from the interesting coupling between FTS and the general DP framework and has been shown to be critical for our algorithms to achieve competitive performances.
> 3. Due to the worst-case nature of our theoretical result, $1-p_t=\mathcal{O}(1/t^2)$ is **a conservative choice** and thus may not be strictly followed in practice. Specifically, in order to derive a theoretical guarantee that is robust against the heterogeneity of agents (i.e., a major challenge in the federated setting), we have upper-bounded **the worst-case error** induced by any set of agents in our theoretical analysis (lines 256-260). The choice of $1-p_t=\mathcal{O}(1/t^2)$ is needed only to ensure that our algorithm is no-regret **even in the worst case** and is hence **a conservative choice**. Therefore, it is reasonable to make $1-p_t$ decrease slower in practice.
> 4. We would like to clarify that it is a commonly accepted practice in Bayesian optimization to set a parameter to a more practical value if the theoretical value is too conservative. For example, the theoretical value for the exploration parameter $\beta_t$ in the classic GP-UCB algorithm has been found to be too conservative by a large number of previous works (see references [a-c] listed below for some examples). Therefore, $\beta_t$ is usually set to more practical (smaller) values in practice.
>
>     In our paper, as we have explained in our response to question 3 above, the choice of $1-p_t=\mathcal{O}(1/t^2)$ is needed only to ensure that our algorithm is no-regret even in the worst case and is hence **a conservative choice**. Therefore, letting $1-p_t$ decrease slower is reasonable in practice. Moreover, as a clarification, for such slowly decreasing $1-p_t$, the regret bound still holds, although it is not no-regret in the worst case.
>
>     Furthermore, we have added an experiment to show that even if we follow the theoretical value of $1-p_t=\mathcal{O}(1/t^2)$ in our experiments, our algorithm is still able to outperform standard TS. See the table below for the validation error after 60 iterations for the human activity recognition experiment (corresponding to Fig. 3 (e)).
>
>    | Method | Validation Error |
>    | :--- | :--- |
>    | TS      | $0.085 \pm 0.0068$ |
>    | FTS-DE ($1-p_t=1/t$) | $0.055 \pm 0.0038$ |
>    | FTS-DE ($1-p_t=1/t^2$) | $0.068 \pm 0.0059$ |
>
>     [a] High Dimensional Bayesian Optimisation and Bandits via Additive Models.
>     [b] Adversarially Robust Optimization with Gaussian Processes.
>     [c] Truncated Variance Reduction: A Unified Approach to Bayesian Optimization and Level-Set Estimation.
>
> 5. Following your suggestion, we have extended Fig. 3 (e) (the human activity recognition experiment) to 200 iterations, and the results are consistent (see table below). The results (validation error) show that after 200 iterations, all methods have converged to very similar values. We will also extend the other experiments to 200 iterations and add the results to the paper after revision.
>
>    | Method | $100$ iterations | $150$ iterations | $200$ iterations |
>    | :--- | :--- | :--- | :--- |
>    | TS      | $0.062 \pm 0.005$ | $0.042 \pm 0.003$ | $0.032 \pm 0.002$ |
>    | FTS-DE | $0.040 \pm 0.003$ |  $0.033 \pm 0.002$ | $0.027 \pm 0.001$ |
>    | DP-FTS-DE ($q=0.35,z=1.0$) | $0.046 \pm 0.003$ | $0.036 \pm 0.002$ | $0.029 \pm 0.002$ |
>    | DP-FTS-DE ($q=0.35,z=2.0$) | $0.050 \pm 0.004$ | $0.037 \pm 0.002$ | $0.028 \pm 0.002$ |
>    | DP-FTS-DE ($q=0.25,z=1.0$) | $0.050 \pm 0.004$ | $0.036 \pm 0.002$ | $0.029 \pm 0.002$ |
>
>
> Minor issues:
> * In Figure 2, the results before the 0th iteration correspond to the initialization period.
> * We will change Line 4 to Line 7 of Algorithm 2 to an if-else statement following your suggestion.
> * You are correct that In Line 84, it should be $\mathbb{R}$. Thank you for pointing this out and we will correct it and double-check all notations.
> * We will change all DNN to DNNs and all RFF to RFFs.
>
> ***
>
> Thank you again for your feedback. We hope that our additional clarifications and experimental results above could improve your evaluation of our paper.

---

> > ### Comment · Reviewer_wuG4 · 2021-08-27
> > **Thank you for your response**
> >
> > Thank you for your response.
> >
> > The table described in A5 is very impressive.
> >
> > However, I would like to ask about the following question again:
> >
> > If $1 - p_t \in \mathcal{O}(1/t^2)$, Line 7 of Algorithm 2 would not be operated with a very high probability.
> >
> > I think this is still a problem where $1- p_t \in \mathcal{O}(1/t)$ or $1 - p_t \in \mathcal{O}(1/\sqrt{t})$ is assumed.
> >
> > Moreover, although Line 5 of Algorithm 2 might work more frequent than Line 7 of Algorithm 2, how does your algorithm work well compared to Thompson sampling?
> >
> > Best regards,
> >
> > Reviewer.

---

> > > ### Author Response · Authors · 2021-08-27
> > > **Further Clarifications**
> > >
> > > Thank you for your additional response.
> > >
> > > To begin with, we would like to emphasize that our algorithm improves the convergence of BO by using information from the other agents (via Line 7 of Algorithm 2) to **accelerate the exploration in the early stage** (lines 54-58), i.e., to help BO quickly identify promising regions in the search space **when few observations are available**. On the other hand, in the long run, our algorithm is expected to perform similarly to standard Thompson sampling (as shown by the table in A5 in our response above) because both algorithms have collected sufficient information (observations) about the agent’s objective function. This is in fact the key point to address your questions:
> > >
> > > > If $1−p_t = \mathcal{O}(1/t^2)$, Line 7 of Algorithm 2 would not be operated with a very high probability. I think this is still a problem where $1−p_t = \mathcal{O}(1/t)$ or $1−p_t = \mathcal{O}(1/\sqrt{t})$ is assumed.
> > >
> > > Since we mainly intend to use information from the other agents (i.e., execute Line 7 of Algorithm 2) **in the early stage**, we only need to ensure that Line 7 of Algorithm 2 is operated with high probability **in the early stage**. This is indeed satisfied because the sequence of $1-p_t$ is chosen to be decreasing and is hence large in the early stage.
> > >
> > > &nbsp;
> > >
> > >
> > > > Moreover, although Line 5 of Algorithm 2 might work more frequent than Line 7 of Algorithm 2, how does your algorithm work well compared to Thompson sampling?
> > >
> > > Because Line 7 of Algorithm 2 is operated frequently (with high probability) **in the early stage**, our algorithm is able to effectively leverage information from the other agents to **accelerate the exploration in the early stage** (as discussed at the beginning of our current response). This is why our algorithm performs better than Thompson sampling as shown in all our experimental results.
> > >
> > > We thank you and will add the discussion here to our revised paper.

---

> > > > ### Comment · Reviewer_wuG4 · 2021-09-01
> > > > **Thank you for clarifications**
> > > >
> > > > Thank you for clarifications.
> > > >
> > > > I increased my score.
> > > >
> > > > I will be happy if you revise the final version of this work by considering other reviewers' feedbacks as well as my concerns.
> > > >
> > > > Best regards,
> > > >
> > > > Reviewer.

---

> > > > > ### Author Response · Authors · 2021-09-02
> > > > > **Thank you and we'll revise the paper accordingly**
> > > > >
> > > > > Thank you for increasing your evaluation of our paper. We will revise the paper carefully to account for your concerns and suggestions, as well as the feedbacks from the other reviewers.
> > > > >
> > > > > Best regards,
> > > > > Authors

---

### Decision · Program_Chairs · 2021-09-28

**Decision:**

Accept (Poster)

**Comment:**

The paper introduces the first federated Bayesian optimisation method with a rigorous privacy guarantee together with theoretical and empirical analysis of its utility. All reviewers recommend acceptance, noting that this is a strong submission.

**Consistency Experiment:**

NeurIPS has a long history of experimentation. In 2014, NeurIPS ran an experiment in which 10% of submissions were reviewed by two independent committees to quantify the randomness in the review process. This year, we repeated a variant of this experiment to see how the quality of the review process has changed over time.  This paper was part of the experiment and was therefore assigned to two committees (consisting of reviewers, an Area Chair, and a Senior Area Chair) that reached independent decisions.  If both committees made the same recommendation, this recommendation was followed. If a single committee recommended acceptance, the paper was accepted (with the exception of a few cases in which the other committee identified what we considered a fatal flaw, e.g., an error in a key result).

This copy’s committee reached the following decision: **Accept (Poster)**

The other committee assigned to the paper recommended **Reject**.  You can find the other set of reviews, along with any follow up discussion with the authors here:
https://openreview.net/forum?id=aj8x18_Te9